# Personalized CRISPR knock-in cytokine gene therapy to remodel the tumor microenvironment and enhance CAR T cell therapy in solid tumors

Michael Launspach [1,2,3,4] ✉, Julia Macos [1,2], Shoaib Afzal [1,2], Janik Hohmann [1], Marc L. Appis [1,2,5], Maximilian Pilgram [1], Stefanie Beez[1], Emily Ohlendorf[1,2], Casper F. T. van der Ven [6], Chahrazad Lachiheb[6], Karin Töws[1,2], Lena Andersch[1,2,3], Marvin Jens [1], Felix Zirngibl [1], Jonas Kath [1,7], Maria Stecklum [8], Elias Rodriguez-Fos [1,9], Kathleen Anders[1,2,3], Dimitrios L. Wagner [4,7,10,11,12,13], Anton G. Henssen [1,3,9,14], Ralf Kühn [14], Angelika Eggert[1,2,3,4,15] & Annette Künkele [1,2,3,4]

The immunosuppressive tumour microenvironment (TME) remains a central barrier to effective immunotherapy in solid tumours. We present a gene-therapeutic strategy that enables localized remodelling of the TME via tumour-intrinsic cytokine expression. Central to this approach is CancerPAM, a multi-omics bioinformatics pipeline that identifies and ranks patient-specific, tumour-exclusive CRISPR-Cas9 knock-in sites with high specificity and integration efficiency. Using neuroblastoma as a model, CancerPAM analysis of tumour sequencing data identifies optimal knock-in sites for pro-inflammatory cytokines (CXCL10, CXCL11, IFNG), and CancerPAM rankings correlate strongly with target-site specificity and knock-in efficiency, validating its predictive performance. CRISPR-mediated CXCL10 knock-in enhances CAR T cell infiltration and antitumour efficacy in vitro and in vivo, including humanized CD34+ HuNOG mice, where CXCL10-expressing tumours show stronger immune infiltration and prolonged tumour control within a reconstituted human immune microenvironment. Our findings establish a framework for safe and effective CRISPR-based cytokine delivery, integrating localized TME remodelling with cellular immunotherapies to enhance CAR T cells and other treatments in immune-refractory solid tumours.

The emergence of CRISPR-Cas9 technology has transformed genome engineering by enabling precise gene editing. Guided by single-guide RNAs (gRNAs), the Cas9 nuclease induces site-specific double-strand breaks, allowing targeted gene knockouts and insertions. This versatility has established CRISPR-Cas9 as an indispensable tool in both genetic research and therapy development[1,2]. Harnessing CRISPR for cancer therapies necessitates highly specific target selection to ensure both safety and efficacy. Automated bioinformatics pipelines integrating multi-omics (genomic, transcriptomic and epigenomic) data have been developed either to advance understanding of cancer biology[3,4] or to facilitate optimal CRISPR gRNA design, to maximize therapeutic precision[5]. However, tools that effectively combine both are still lacking.

**Fig. 1 | Therapeutic concept.** Schematic of an augmentative therapeutic approach combining the CancerPAM multi-omics-based target identification pipeline with CRISPR/Cas9-targeted cytokine gene therapy, followed by CAR T cell or other immunotherapies. CancerPAM enables the identification of promising tumour-specific CRISPR knock-in target sites in solid tumours with an immunosuppressive microenvironment. At these sites, cytokine transgenes are integrated, leading to beneficial tumour biology changes that enhance subsequent CAR T cell or other immunotherapies. **Illustration attribution:** Schematic components: Created in BioRender. Kuenkele, A. (2025) https://BioRender.com/u71d212.

Despite advances in precision oncology, solid tumours remain a major global health burden[6]. Directly targeting tumour cells through cytotoxic gene therapy is hindered by the necessity to deliver the therapy to every cancer cell to prevent relapse, a requirement current delivery vehicles fail to meet[7–9]. Immunotherapies have emerged as a promising alternative, leveraging the immune system's ability to recognize and eliminate tumours. Oncolytic viruses, neoantigen-based vaccines and chimeric antigen receptor (CAR) T cell therapy have shown considerable potential[10–12]. CAR T cells, genetically modified to express synthetic receptors that combine tumour antigen recognition with intracellular signalling domains, directly recognize target antigens and mount a cytotoxic response[13]. However, the immunosuppressive tumour microenvironment (TME) in many solid tumours presents a significant hurdle to CAR T cell therapy. Tumours deploy various immune evasion mechanisms, including secretion of immunosuppressive cytokines and recruitment of regulatory T cells, which collectively hinder CAR T cell infiltration and cytotoxic activity[14,15].

Neuroblastoma is a paediatric solid tumour of neuroectodermal origin that has an immunosuppressive TME. Despite aggressive multimodal treatment regimens, high-risk neuroblastoma is associated with poor long-term survival[16,17]. CAR T cell therapies targeting antigens on neuroblastoma cells, such as L1CAM, or GD2 have been explored[18,19]. While CAR T cell therapy against GD2 has shown some promise (NCT03373097), other clinical trials, including those targeting L1CAM (NCT02311621), have reported limited efficacy due to antigen escape and poor infiltration and persistence in the hostile TME[20–22]. This immune exclusion is particularly pronounced in MYCN-amplified tumours (~25% of neuroblastomas), which exhibit transcriptional repression of interferon-stimulated genes, impaired antigen presentation, and reduced infiltration of CD8+ T cells and NK cells[23–25]. Low IFNG signatures and diminished chemokine expression, especially T cell-attracting chemokines (e.g. CXCL10), correlate with poor prognosis and impaired immune responsiveness in neuroblastoma[23,24,26]. Conversely, tumours with robust T cell-inflamed gene expression signatures that include IFNG and its downstream chemokines are associated with improved overall survival[27]. Together, these findings underscore the central role of the interferon-chemokine axis in shaping the neuroblastoma TME.

Based on this evidence, we select CXCL10, CXCL11, and IFNG as rational proof-of-concept candidates for proposing a gene-therapeutic approach that remodels the TME by inducing tumour-intrinsic expression of immunostimulatory cytokines. CXCL10 and CXCL11, ligands of CXCR3 expressed on activated CD8+ T cells, NK cells, and CAR T cells, potently enhance immune infiltration and have been linked to improved immunotherapy outcomes across multiple models and entities[28–31]. IFNG, a master regulator of antitumour immunity, not only induces these chemokines but also upregulates MHC class I and antigen presentation machinery, thereby orchestrating multiple arms of immune activation[24,32,33]. To implement this concept with precision, we develop CancerPAM - an integrative multi-omics pipeline that identifies and ranks tumour-specific CRISPR knock-in sites based on specificity, efficiency, and safety criteria. Using neuroblastoma as a model, we demonstrate that CancerPAM enables efficient, site-specific integration of cytokine transgenes into tumour genomes to improve CAR T cell infiltration and efficacy (Fig. 1). Functional validation in vitro and in vivo confirms that tumour-intrinsic CXCL10 expression enhances CAR T cell infiltration and improves tumour control. These findings highlight tumour-intrinsic cytokine expression as a promising strategy to enhance solid tumour immunotherapy and establish CancerPAM as a powerful tool for CRISPR-based precision interventions.

## Results

### The CancerPAM pipeline identifies and ranks tumour-specific novel PAM sites

Identification of tumour-specific CRISPR target sites is crucial to develop precise and efficient gene-editing strategies. We established a manual step-by-step process to identify and select single-nucleotide variants (SNVs) in tumour sequencing data that form novel 5'-NGG-3' sequences, where N is any base, as protospacer adjacent motifs (PAM) recognized by the Cas9 nuclease (for simplicity, we refer to this specific type as *PAM site* throughout the manuscript) to effectively cleave target sequences in tumours. We tested this manual process on whole-exome sequencing (WES) data from neuroblastoma cell lines and identified promising novel PAM sites for subsequent knock-in experiments (Supplementary Fig. 1a, and Supplementary Table 1). To automate and improve this selection process, we developed CancerPAM. CancerPAM is a Python-based modular pipeline that integrates WES - or whole-genome sequencing (WGS) data with multi-omics annotation to identify and rank novel PAM sites (Fig. 2a). The corresponding gRNA sequence is automatically determined, then the gene expression, copy number, gene dependency, Doench[5] and Moreno[34] CRISPR cutting efficiency scores, cutting frequency determination (CFD)[5] and Massachusetts Institute of Technology (MIT)[35] specificity scores are annotated. By incorporating these key biological and CRISPR-associated parameters, CancerPAM prioritizes optimal editing sites using both feasibility and safety criteria in a weighted ranking algorithm

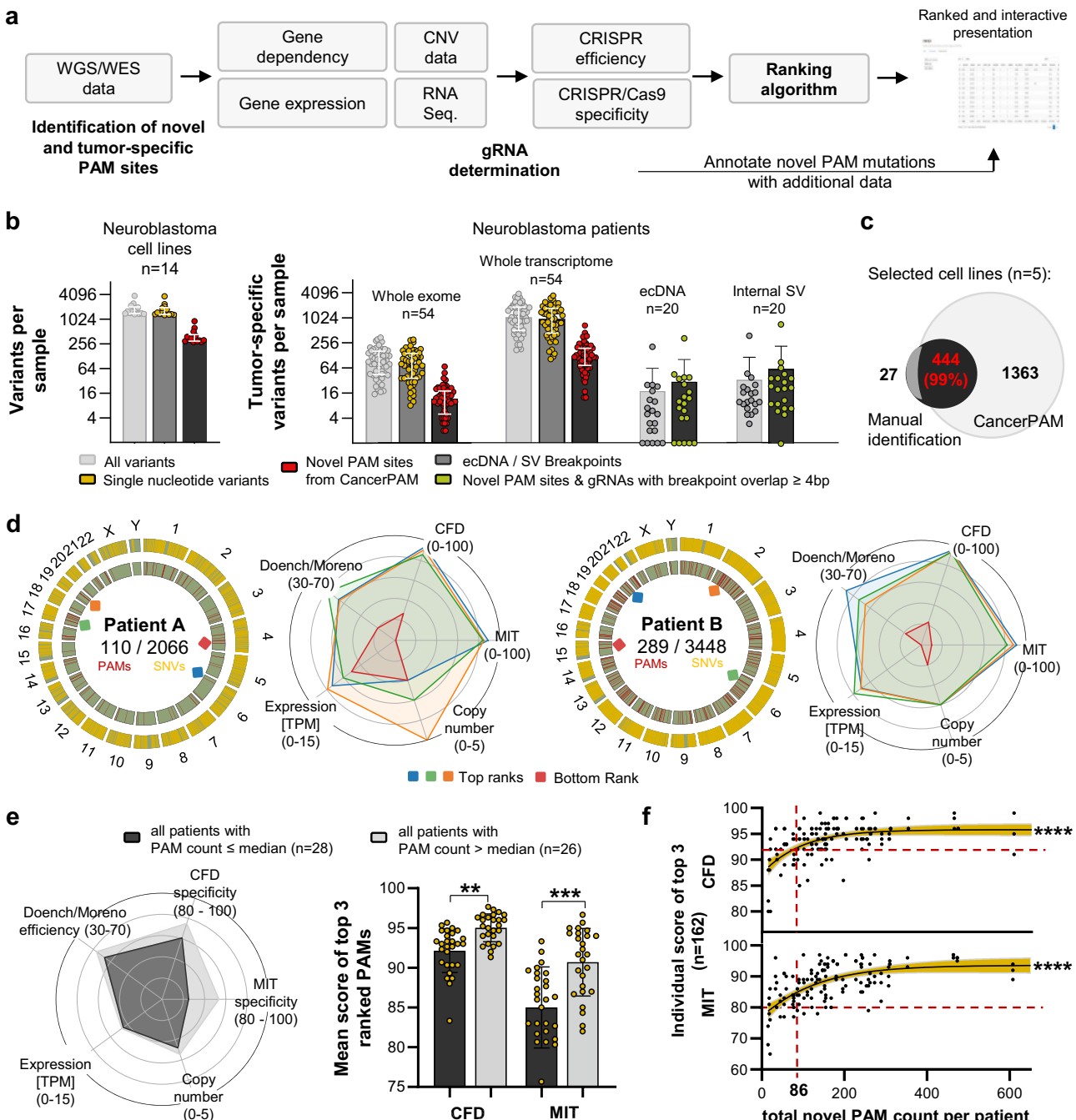

**Fig. 2 | The CancerPAM multi-omics-based automated pipeline identifies and ranks targetable tumour-specific PAM sites in cancer cell lines and patient samples. a** Overview of CancerPAM integrating whole-genome/whole-exome sequencing (WGS/WES) data with multi-omics annotation to identify and rank novel PAMs. **b** Quantification of variants and tumour-specific PAMs in neuroblastoma cell lines (WES) and patients (WGS including extrachromosomal DNA (ecDNA)). "All variants" refers to single-nucleotide variants (SNVs) and small insertions/deletions (InDels) (<300 bp) from the CancerPAM pipeline. "Genomic structural variants (SVs)" = larger rearrangements (deletions, insertions, tandem duplications, inversions, translocations; SV/ecDNA pipeline). **c** Overlap between novel PAM sites identified via manual screening and those identified using the CancerPAM pipeline. **d** Chromosomal distribution of SNVs and novel tumour-specific PAM sites and visualization of annotated feature characteristics for the top three highest- and lowest-ranked PAM site in two patients (A and B) including expression of the gene containing the PAM in transcripts per

million (TPM). **e** Mean annotated feature values of the top three ranked PAM sites, averaged across patients with a high (>median; $n = 26$) or low (≤median; $n = 28$) PAM count. Statistical comparison of these two groups is shown for the mean cutting frequency determination (CFD) and Massachusetts Institute of Technology (MIT) specificity scores. **f** Correlation analysis of individual scores for the top three ranked novel PAM sites across patients ($n = 162$ individual PAM sites, top three per patient) versus total tumour-specific PAM count per patient. Red dashed lines indicate potential safety and feasibility thresholds. Data presentation: **b**, **e** Mean ± SD. Dots and $n$ represent individual cell lines or patients. Statistical tests: **e** Kruskal–Wallis with Dunn's post-hoc correction (two-sided, exact $p = 0.0071$, 0.0006); **f** Logistic regression for curve fitting with error bands representing the 95% confidence interval followed by two-sided Spearman correlation analysis (exact $p < 0.0001$, <0.0001). $p$ values: **"<0.01, ***<0.001, ****<0.0001; *n.s.*, not significant. Source data are provided as a Source Data file.

(Supplementary Fig. 1b). We hypothesised that several factors contribute to novel PAM site optimality, beyond gRNA sequence-based CRISPR efficiency scores. Specifically, we posited that higher gene expression, which correlates with chromatin accessibility[36], and increased copy number would enhance knock-in efficiency. High specificity scores would reduce off-target activity, thereby improving safety, while low gene dependency would mitigate potential selection disadvantages. Higher expression levels of the target are also expected to reduce the risk of CRISPR editing causing pro-tumourigenic effects, such as inadvertently targeting a functional tumour suppressor gene - a risk already mitigated by the presence of the selected PAM-creating mutation, which may itself impair gene function. To implement these considerations, we applied a ranking algorithm that weighted the CFD specificity score twice to enhance safety. All other feature values are considered equally for ranking, except for the Doench and Moreno CRISPR cutting efficiency scores, which, given their limited predictive accuracy[37], are compiled as a single predictor value (mean of both scores). Applying CancerPAM to neuroblastoma datasets, we identified a substantial number of novel PAM sites in both neuroblastoma cell lines and tumour samples from patients (matched tumour/healthy tissue data from diagnosis). In 14 neuroblastoma cell lines, a median of 327 novel PAM sites were identified from 1470 SNVs (23% [95% CI: 22–24%]) in exonic regions. In tumour data from 54 patients, a median of 12 novel PAMs were identified from 82 SNVs (17% [95% CI: 13–22%]) in exonic regions, and 130 novel PAMs from 1190 SNVs (14% [95% CI: 12–15%]) across the entire transcriptome (Fig. 2b, and Supplementary Note 1). To evaluate CancerPAM pipeline accuracy and sensitivity, we conducted manual curation of findings for five neuroblastoma cell lines and cross-validated CancerPAM predictions. The pipeline achieved 99% concordance with manually curated novel PAM sites from whole-exome data, confirming its reliability (Fig. 2c). Consistent with exonic data analysis, CancerPAM results revealed that tumour-specific PAMs were unevenly distributed across the genome, clustering in gene-dense regions (i.e. chromosomes 11, 17 and 19). To complement these SNV-derived findings, we performed an additional extra-chromosomal DNA (ecDNA)-specific analysis using WGS data from 20 neuroblastoma tumours[38]. Since ecDNA breakpoints generate novel sequence junctions absent from the reference genome, we developed a dedicated analysis code (outside CancerPAM) to systematically screen such junctions. This revealed both novel PAM sequences directly created at breakpoint sites and gRNA candidates spanning the junction, where we required at least four nucleotides of complementarity across the breakpoint to increase the chance of tumour-specific recognition. Across 20 patients, we identified a median of 10 novel PAMs and gRNAs per sample at a median of 5 ecDNA breakpoints, and a median of 22 novel PAMs and gRNAs per sample at a median of 11 structural variant breakpoints, respectively. Together, these analyses identified a distinct but comparatively small set of tumour-specific CRISPR target sites, thereby expanding the mutational landscape considered by CancerPAM (Fig. 2b). Across all patient samples, 8750 novel PAMs were identified from SNVs in 56 individuals. Of these, 50 PAMs were observed in two patients, 4 PAMs in three patients, and 2 PAMs in four patients, corresponding to an overall recurrence rate of 1.4% across the cohort. In parallel, analysis of structural variants and ecDNA junctions in 20 patients revealed 1818 novel PAMs and breakpoint-spanning gRNAs in total, of which 16 PAMs and 35 gRNAs were re-detected in a second patient. This corresponds to a recurrence rate of 5.6%. Taken together, these data demonstrate that recurrence of tumour-specific PAMs is rare both for SNV-derived and structural variant (SV)/ecDNA-derived sites, with consistently low frequencies across patients, underscoring the strong inter-patient heterogeneity of neuroblastoma and supporting individualized targeting strategies such as CancerPAM (Fig. 2b, Supplementary Figs. 1d and 2f, and Supplementary Notes 2 and 3). CancerPAM accurately annotated key feature values, allowing data-driven selection of

the most viable CRISPR target sites (Fig. 2d, Supplementary Fig. 5a–g, and Supplementary Table 2). To validate the weighted ranking algorithm, we assessed correlations between PAM rank and various biological and CRISPR-associated features. Higher PAM rank showed strong positive correlation with CRISPR specificity scores and moderate correlation with CRISPR efficiency scores and gene dependency (Supplementary Fig. 6a–g), but did not correlate with gene expression or copy number. Tumour sample groups bearing many or few PAM sites (above and below median) were compared regarding annotated features. Top-ranked PAM sites in tumour genomes harbouring many PAM sites exhibited significantly higher specificity scores, that correlated strongly with the total PAM site count in these tumour genomes (Fig. 2e). We determined a potential feasibility and safety threshold using the total novel PAM sites detected in a tumour sample. Tumour samples with at least 86 novel PAM sites (72% [39/54] of the cohort) had a > 90% probability that their top three ranked PAM sites possessed a CFD specificity score >90 and MIT specificity score >80 (Fig. 2f, and Supplementary Notes 4 and 5). To sum up, we identified numerous tumour-specific novel PAM sites from both SNVs and, to a lesser extent, from structural variant and ecDNA breakpoints. CancerPAM achieves 99% accuracy in detecting these sites and systematically ranks them according to feasibility and safety for CRISPR target prioritization.

## CRISPR-mediated knock-in of cytokine transgenes at tumour-specific target sites is feasible and efficient

We selected two neuroblastoma cell lines (SK-N-BE2c: *MYCN*-amplified, SK-N-AS lacking *MYCN* amplification) in which to evaluate the feasibility of site-specific cytokine transgene integration. The most promising 9 novel PAM sites identified during manual selection were chosen, and also identified by CancerPAM (Fig. 3a, Supplementary Fig. 1a, and Supplementary Table 1). Sanger sequencing confirmed the presence of the selected novel PAM sites in 7 cases, while also verifying target site absence in the respective cell line where it was not present in WES data. Mutation frequencies in the respective cell line (harbouring the novel PAM site) calculated from conformational Sanger sequencing varied between 0 and 20% for novel PAM sites located in *SH3BP1* and *SNX18*; 30–60% for *CHD1, RBM12, SCAF11, AP1M1* and *CHST11*; 66% for *RPLPO* (2 of 3 alleles) and 100% for *IGSF9B* (2 of 2 alleles) (Fig. 3c). These novel PAM sites were targeted with specific Cas9/gRNA ribonucleoproteins (RNP) (Supplementary Note 6) while co-delivering homology-directed repair template (HDRT) for CXCL10, CXCL11 or IFNG transgene knock-in (Fig. 3b, and Supplementary Fig. 9c, d). Cytokine transgenes contained a custom-designed EF1α-derived shortened promoter and Q8 epitope tag for flow cytometric detection (Supplementary Fig. 10a, and Supplementary Note 7). Successful knock-in at the target sites was flow cytometrically confirmed 28 days after RNP/HDRT transfer (Fig. 3d), and was most efficient at the *IGSF9B* locus in SK-N-BE2c and *RPLPO* locus in SK-N-AS (Supplementary Note 8). With the exception of the *IGSF9B* locus (knock-in occurred in both cell lines), gRNAs facilitated relevant knock-in only in cell lines harbouring the novel PAM mutation (Fig. 3d). An alternative PAM site was present adjacent to the novel PAM site in *IGSF9B* in the SK-N-AS cell line, which lacks the novel *IGSF9B* PAM site, potentially explaining the unexpected integration events. Knock-in efficiency strongly positively correlated with copy number and gene expression, and weakly positively correlated with Doench and Moreno CRISPR efficiency scores (Fig. 3e). Site-specific cytokine transgene knock-in was validated by digital PCR. Specificity was confirmed and quantified using In/In and Out/In fluorescent probe-based assays that distinguish precise knock-in from random integrations or free-floating DNA. The results demonstrated robust integration of CXCL10, CXCL11 and IFNG at target loci (Fig. 4a–c, Supplementary Fig. 12a, and Supplementary Note 9). IFNG, however, was integrated at significantly lower rates (Fig. 4d), in line with flow cytometry findings 28 days after knock-in

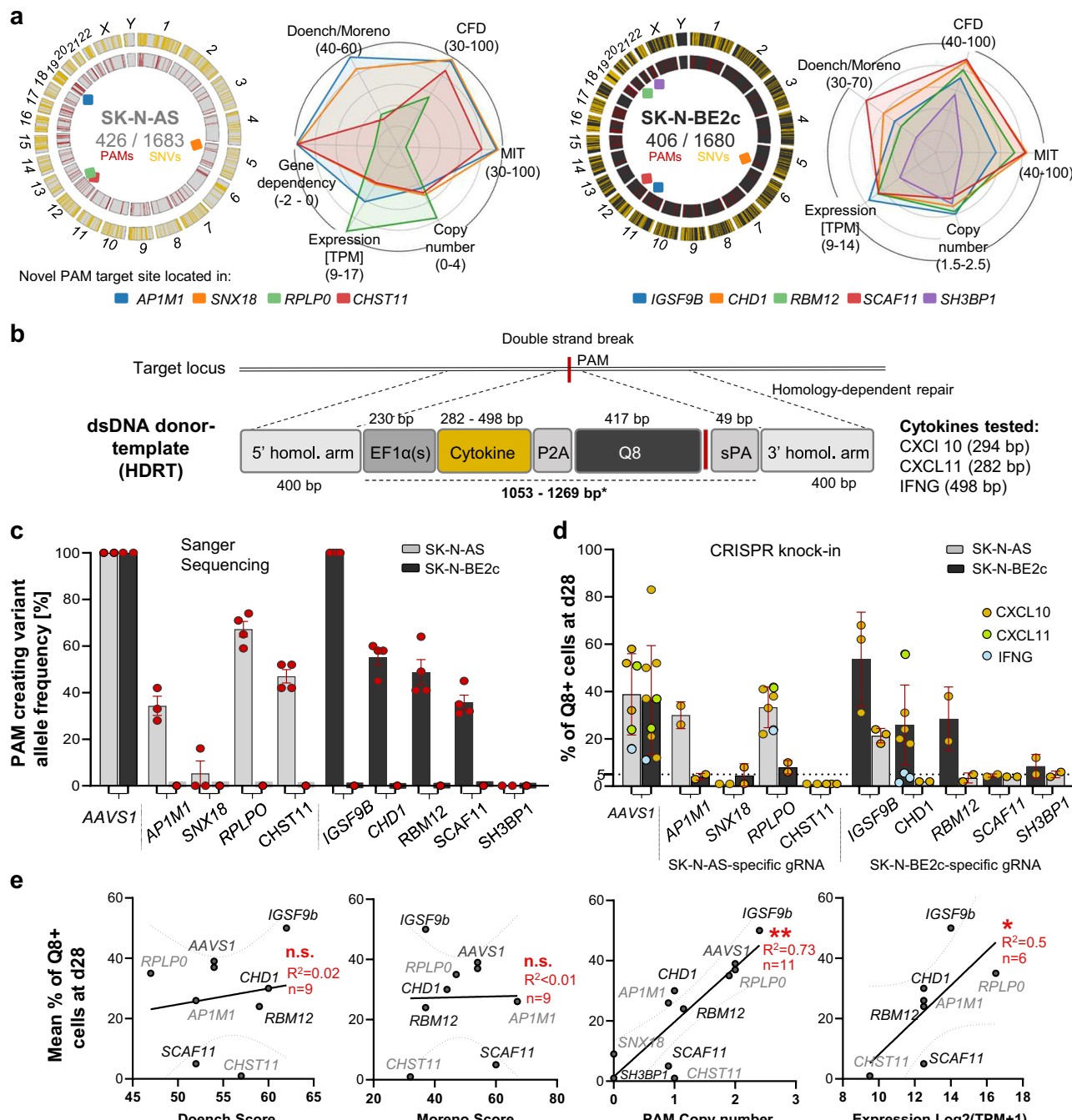

**Fig. 3 | Gene-therapeutic CRISPR knock-in of cytokine transgenes is efficient and specific for top-ranked novel PAM targets. a** Circos plots showing the chromosomal distribution of single-nucleotide variants (SNVs, yellow) and novel PAM sites (red) in SK-N-BE2c and SK-N-AS neuroblastoma cell lines. PAMs tested in knock-in experiments are marked with coloured squares. Radar plots visualize annotated features for these PAMs, named after their respective genes. **b** Schematic of CRISPR/Cas9-mediated transgene knock-in via homology-directed repair (HDR) following ribonucleoprotein (RNP) electroporation. The linear double-stranded (ds) homology-directed repair donor template (HDRT) consists of 5' and 3' homologous arms, 400 base pairs (bp) each, a custom EF1α-derived promoter, the cytokine transgene linked by a P2A self-cleaving peptide to a stainable Q8 reporter (CD34 epitope, CD8 transmembrane domain), followed by a stop codon and synthetic poly(A) (sPA) sequence. **c** PAM-creating mutation allele

frequency analysed by Sanger sequencing of selected targets in SK-N-BE2c and SK-N-AS. **d** Knock-in efficiency of three cytokines (CXCL10, CXCL11, IFNG) at different target sites, measured by Q8 antigen expression by flow cytometry at day 28 post-electroporation. **e** Correlation of mean knock-in rate (Q8 antigen expression at day 28) with PAM-annotated features, including CRISPR efficiency scores (Doench and Moreno), PAM copy number, and expression of the gene containing the PAM for selected targets. Each dot and *n* represent the mean value for the selected locus-cell line combinations from (**d**). Data presentation: **c, d** Mean ± SD. Each dot represents one biological replicate and exact *n* values per condition are provided in the Source Data file. Statistical analysis: **e** Linear regression for curve fitting with $R^2$ values and error bands representing the 95% confidence interval followed by two-sided Spearman correlation analysis (exact *p* = 0.62, 0.82, 0.0022, 0.019). *p* values: *<0.05, **<0.01; *n.s.*, not significant. Source data are provided as a Source Data file.

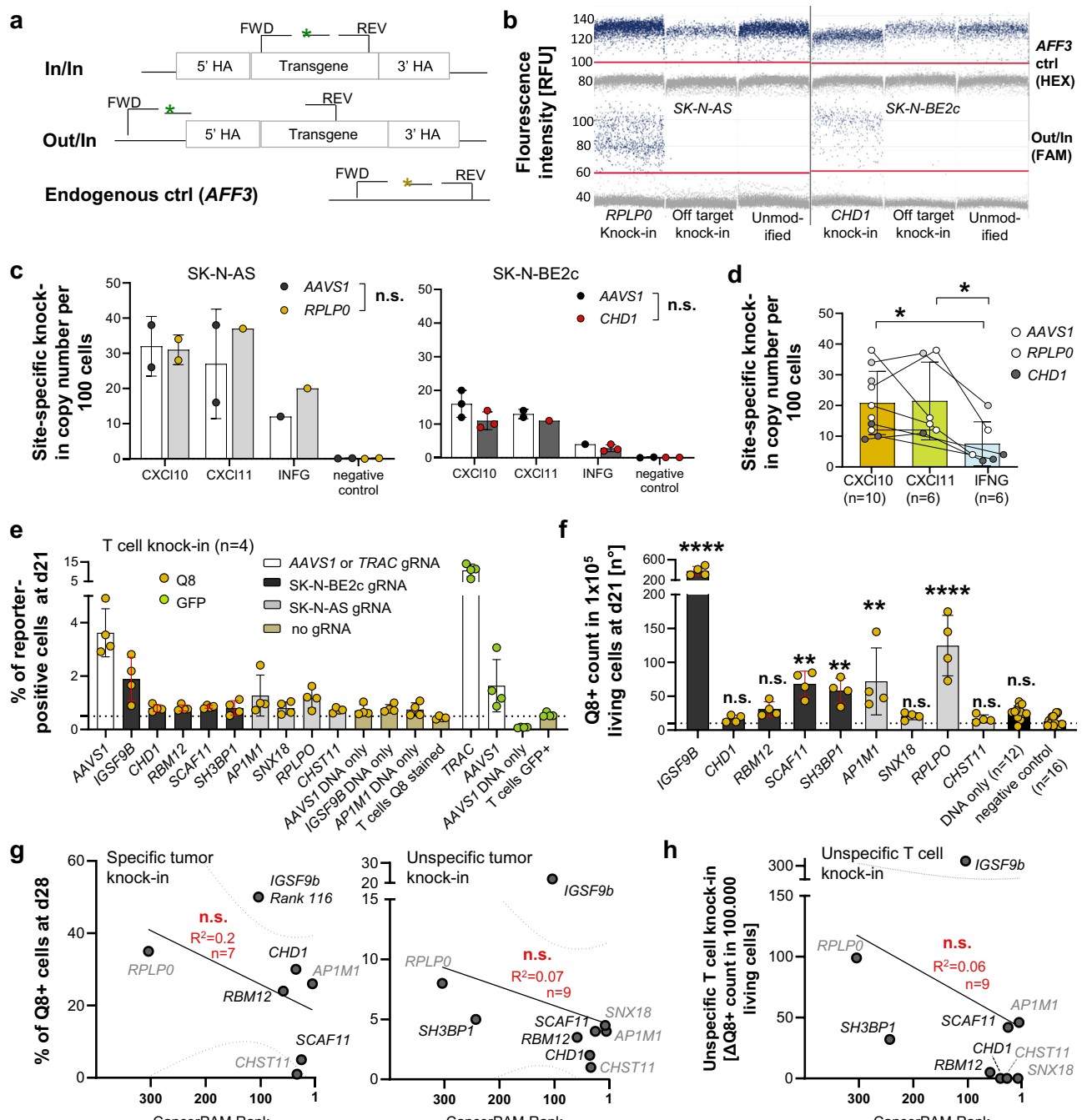

**Fig. 4 | The CancerPAM pipeline accurately identifies novel PAM sites with a low unspecific knock-in risk. a** Schematic of digital PCR assays for site-specific CRISPR knock-in confirmation. "In/In" refers to a fluorescence probe-based PCR assay with primers binding inside the transgene, while "Out/In" uses a forward primer upstream of the homologous arm. A probe-based assay for the *AFF3* gene served as an endogenous control. **b** Digital PCR raw data showing positive partitions (blue) for knock-in and control samples using the Out/In or *AFF3* control assay. Representative of biological replicates shown in (**d**), with similar results obtained in all replicates. **c** Site-specific knock-in copy number per 100 cells, 7 days after RNP/HDRT electroporation in SK-N-AS and SK-N-BE2c for different transgenes and loci. **d** Cumulative site-specific knock-in copy numbers for CXCL10, CXCL11, and IFNG. **e** Comparison of reporter (Q8/GFP) positive cells 21 days post-CRISPR knock-in in primary T cells from two donors using gRNAs targeting neuroblastoma-specific novel PAM sites or AAVS1 and TRAC controls ($n = 4$, 2 biological replicates per donor). **f** High-throughput flow cytometry (>300,000 cells analysed per sample) to compare unspecific knock-

in rates for Q8-reporter HDRTs at neuroblastoma-specific PAM sites after T cell knock-in ($n = 4$). **g** Correlation of specific knock-in rates (in novel PAM-harbouring neuroblastoma cells) and unspecific knock-in rates (in cells lacking the novel PAM) with CancerPAM-assigned ranks. **h** Correlation of unspecific CRISPR/Cas9-mediated T cell knock-in rates, as determined in (**f**), with CancerPAM ranks. Data presentation: **c**–**f** Mean ± SD. Each dot represents one biological replicate. $n$ refers to the number of biological replicates and exact $n$ values are provided in the Source Data file. **g**, **h** Dots and $n$ represent the mean value for the selected locus-cell line combinations from Fig. 3d. Statistical analysis: **c**, **d** Two-way ANOVA with Tukey test (two-sided, exact $p = 0.42$, 0.14, 0.01, 0.025); **f** Kruskal–Wallis with Dunn's post hoc test (two-sided, exact $p = <0.0001$, >0.99, 0.38, 0.002, 0.0076, 0.0058, >0.99, 0.0004, >0.99, 0.12); **g**, **h** Linear regression for curve fitting with error bands representing the 95% confidence interval followed by two-sided Spearman correlation analysis (exact $p = 0.14$, 0.21, 0.35). $p$ values: \*<0.05, \*\*<0.01, \*\*\*<0.001, \*\*\*\*<0.0001; n.s., not significant. Source data are provided as a Source Data file.

(Supplementary Fig. 10c). Culturing untreated SK-N-BE2c cells with IFNG showed signs of IFNG-mediated toxicity and growth impairment (Supplementary Fig. 12d), which could indicate early cell loss following IFNG knock-in (before DNA isolation for digital PCR) or impairment of homology-directed repair (HDR). To further validate pipeline ranking and CRISPR specificity score predictive value, we performed a knock-in experiment using primary T cells isolated from two healthy donors and the same gRNA set. High-throughput flow cytometry demonstrated low but significant unspecific knock-in for gRNAs targeting the *IGSF9B*, *SCAF11, SH3BP1, AP1M1* and *RPLP0* sites. No unspecific knock-in was observed for gRNAs targeting the *CHD1, RBM12, SNX18* and *CHST11* sites. (Fig. 4e, f, Supplementary Fig. 13a, and Supplementary Note 10). However, only unspecific knock-in in the cases of *IGSF9B* and *RPLP0* resulted in CXCL10 levels exceeding variable baseline levels (Supplementary Fig. 13b). While not statistically significant, correlation analysis revealed a trend toward inverse correlation between unspecific knock-in rates and CancerPAM ranking scores. The highest-ranked PAM sites demonstrated the lowest frequency of off-target integration, further supporting the utility of the CancerPAM algorithm in prioritizing safe and effective gene-editing targets (Fig. 4g, h). To further assess safety, we also estimated the potential statistical risk of tumour-specific off-target sites, which correlated inversely with CancerPAM with 95% of top-ranked gRNAs (ranks 1–3) showing a calculated risk of <0.01%. (Supplementary Fig. 13e). In conclusion, successful CRISPR-mediated site-specific cytokine transgene integration was achieved and confirmed by digital PCR. Knock-in efficiency correlated strongly with gene expression and copy number, with minimal off-target effects. CancerPAM ranking was associated with reduced unspecific knock-in, validating its predictive accuracy for safe gene editing. Novel tumour-specific PAM site discovery and ranking by CancerPAM provide optimal support for experimental design and efficient CRISPR-mediated cytokine transgene integration.

## Tumour cytokine secretion achieved by CRISPR-mediated knock-in improves CAR T cell infiltration and efficacy in vitro and in vivo

To investigate the effects of cytokine secretion on CAR T cells, we enriched transgenic neuroblastoma cell lines using fluorescence-associated cell sorting for PE-labelled Q8-reporter-positive cells. Flow cytometry and digital PCR (dPCR) confirmed stable transgene expression and site-specific knock-in, except in the case of IFNG into SK-N-BE2c (Fig. 5a, Supplementary Fig. 14, and Supplementary Note 11). While unmodified cell lines did not show any CXCL10, CXCL11 or IFNG expression or secretion, elevated levels of these cytokines were confirmed in the knock-in cell lines by ELISA-based quantification (Fig. 5b, and Supplementary Fig. 34). Tumour-intrinsic IFNG expression upregulated HLA-ABC and increased FAS and PD-L1 in three out of four conditions, with similar effects in pre-sort and enriched cells, indicating that even low expression levels can trigger tumour-wide marker upregulation. In parallel, tumour-secreted IFNG significantly inhibited M2-like macrophage polarization with comparable effects between *AAVS1* and *RPLP0* targeting, but a tendency toward dose dependency when comparing pre-sort to post-enrichment conditions (Fig. 5c, and Supplementary Fig. 26). CAR T cells demonstrated significantly better tumour growth control over 96 h compared with non-transduced cells (effector-to-target ratio, 1:5, cytokine-expressing and enriched SK-N-AS neuroblastoma cells after *AAVS1* locus knock-in co-cultured with L1CAM-targeting CAR T cells; Fig. 5d). However, complete tumour cell eradication was not achieved under any condition. Cytotoxicity in IFNG-expressing tumour cells was more pronounced at 24 h, but no longer statistically significant by 72 h (Fig. 5d, Supplementary Fig. 16a, and Supplementary Note 12). To directly compare general safe-harbour with tumour-specific knock-in, we performed co-culture experiments using IFNG-expressing SK-N-AS cells with integrations at *AAVS1* or the tumour-specific *RPLP0* locus. Both pre-sort and enriched

conditions showed significantly enhanced killing within the first 24 h, but no differences between *AAVS1* and *RPLP0* (Supplementary Fig. 27a). Even non-enriched knock-in cells demonstrated this effect, indicating that low-frequency integration can still have measurable biological consequences. In SK-N-BE2c cells, stable IFNG knock-in could not be achieved; dose-dependency was therefore modelled by adding recombinant IFNG at varying concentrations. In this setting, no additional IFNG-mediated effect on killing was detected, likely because high L1CAM expression caused rapid CAR T cell killing that masked subtle cytokine effects (Supplementary Fig. 27b). Conditioned media from CXCL10- and CXCL11-expressing tumour cells significantly enhanced CAR T cell migration compared with media from control tumour cells using two distinct transwell migration assays (Supplementary Fig. 18a–e). In vitro 3D bioprinted tumour models[39] were used to further investigate transendothelial migration and tumour infiltration. A substantial increase in CAR T cell tumour infiltration into CXCL10- and CXCL11-expressing tumours, both in the presence and absence of an additional endothelial layer was confirmed in 12 h infiltration experiments, after relevant CAR T cell proliferation (measured by Ki-67 positivity) within 12 h post-co-culture initiation had been excluded as a confounder in a preliminary experiment (Fig. 5e, f, and Supplementary Fig. 19c). We conclude that tumour-secreted CXCL10 and CXCL11 enhance CAR T cell migration.

In vivo luciferase-expressing L1CAM-targeting CAR T cell trafficking and expansion was monitored by longitudinal bioluminescence imaging (BLI) in mice subcutaneously xenografted with either transgenic or unmodified tumour cell lines in the flank. Tumours with low (SK-N-AS) or high (SK-N-BE2c) L1CAM antigen expression were evaluated (Fig. 6a, Supplementary Figs. 20–23, and Supplementary Note 13). Overall tumour-localized CAR T cell mass - including infiltrated and expanded T cells - was quantified as the flank BLI area under the curve (AUC) from day 1 until the final BLI measurement prior to sacrifice. A positive trend in tumour-localized CAR T cell mass was observed for mice bearing cytokine-expressing tumours compared with those with unmodified tumours (+37% in SK-N-AS and +18% in SK-N-BE2c), although this did not reach statistical significance. Notably, mice bearing cytokine-expressing tumours exhibited significantly higher BLI signals at day 4 post-injection, indicative of enhanced CAR T cell infiltration (SK-N-AS: +221%, *p* = 0.026; SK-N-BE2c: +132%, *p* = 0.038; Fig. 6b, c, and Supplementary Fig. 24c, d). In terms of therapeutic efficacy, no tumour remission was observed in mice bearing L1CAM-low (SK-N-AS) tumours, regardless of CXCL10 expression (Fig. 6d). CAR T cell efficacy and tumour control in mice bearing L1CAM-high (SK-N-BE2c) tumours, however, varied across treatment groups. L1CAM-knockout tumours rapidly grew by day 11 (median survival: 7 days), while partial remission occurred in three animals bearing unmodified tumours, (assessed until day 22, median survival: 11 days; Fig. 6d, and Supplementary Fig. 24e). Regression was enhanced for CXCL10-secreting tumours, with three animals being tumour-free by day 22, then relapsing after day 26, as the CAR T cell BLI signal disappeared (median survival: 14 days, i.e. +27% compared with unmodified tumour-harbouring animals; Fig. 6d, Supplementary Figs. 20a, b, e, and Supplementary Note 14). To further assess the in vivo relevance of tumour-intrinsic CXCL10 expression under immune-competent conditions, we employed humanized NOG (HuNOG) mice engrafted with human CD34[+] haematopoietic stem cells and subsequently xenografted with SK-N-BE2c neuroblastoma cells with or without CXCL10 knock-in at the *CHD1* locus. Immune reconstitution was confirmed by flow cytometry 140 days post CD34[+] stem cell transplantation (Fig. 7a, c). Following tumour implantation, both groups showed comparable engraftment efficiency and initial tumour sizes (Fig. 7b). CXCL10-expressing tumours displayed significantly elevated CXCL10 serum concentrations both before CAR T cell infusion and at the study endpoint (Fig. 7d). Mice were eligible for efficacy analysis if they exhibited measurable flank tumours and no

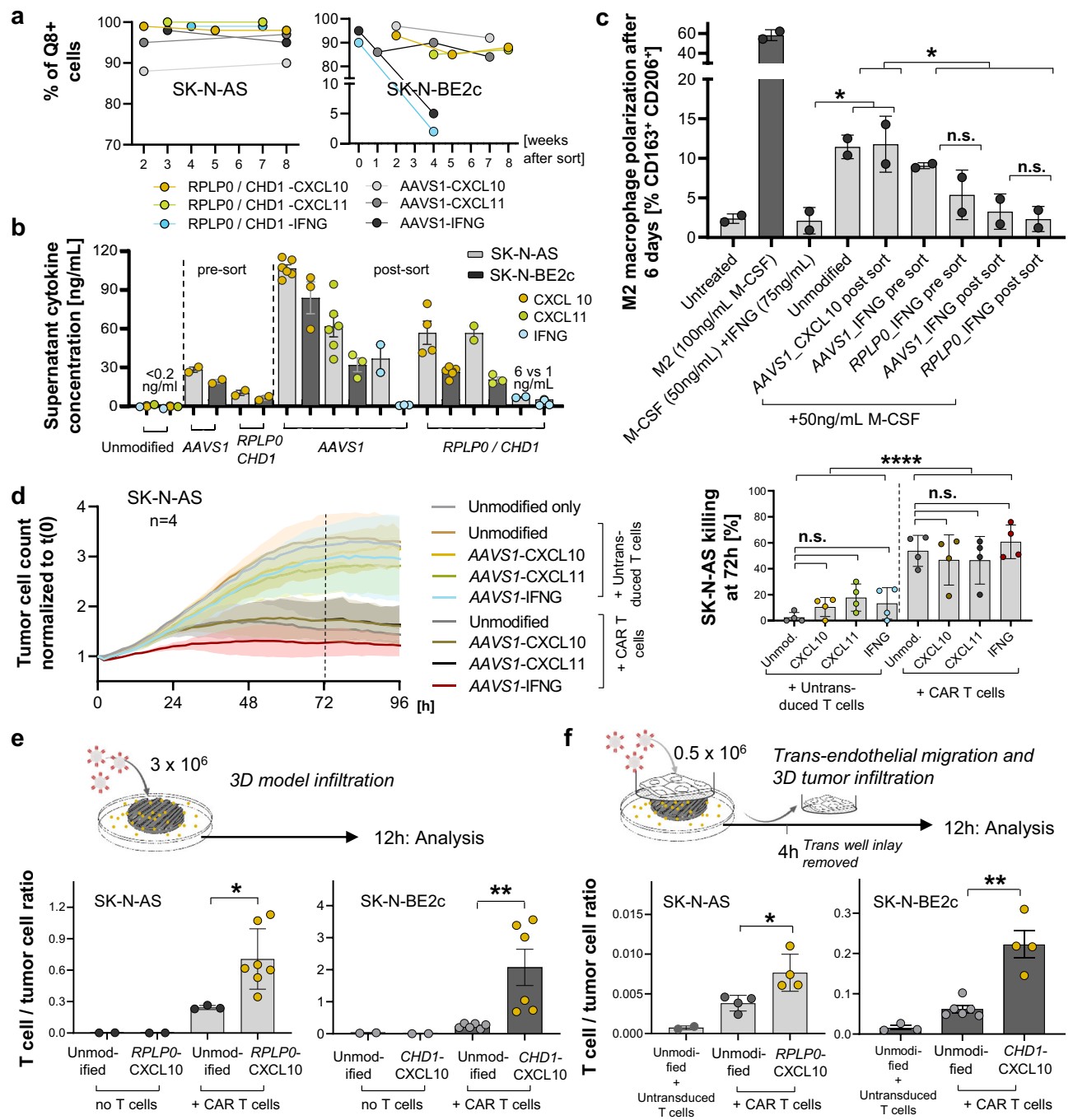

clinical signs of GvHD. Four animals were excluded from the primary analysis (two CXCL10, two unmodified) due to signs of pre-infusion GvHD ($n = 3$) or ectopic tumour localisation in the ovary ($n = 1$). CXCL10-expressing tumours demonstrated significantly slower growth kinetics, higher tumour control rates at day 14 (88% vs. 29%, $p = 0.04$), and prolonged median event-free survival (49 vs. 21 days; $p = 0.0009$; HR = 4.7, 95% CI 1.2–17.9) compared with controls (Fig. 7e–g, and Supplementary Fig. 30a). An intention-to-treat sensitivity analysis, including excluded mice as non-controlled, confirmed the survival benefit ($p = 0.005$; HR = 3.2; 95% CI 1.1–9.3; Supplementary Fig. 30b–d). Notably, one animal achieved complete remission, with no detectable tumour relapse until day 56. Having received autologous CAR T cells derived from its own donor pool indicates full CAR T cell

functionality and compatibility within the reconstituted human immune system. Longitudinal BLI did not resolve CAR T cell tumour infiltration in this model, likely due to endogenous immune-cell competition and reduced signal-to-noise ratios in the reconstituted immune setting (Supplementary Fig. 33a–c). Nevertheless, functional BLI controls and endpoint flow cytometry confirmed CAR T cell presence in circulation and tumours, excluding early complete CAR T cell rejection (Fig. 7h, i, Supplementary Fig. 33a, d and and Supplementary Note 15). Circulating CAR T cells peaked 2 days post infusion and subsequently declined to baseline levels (Fig. 7h). CXCL10-expressing tumour-bearing mice showed a significant decrease in peripheral human leukocyte proportions between day 28 and 56, resulting in a potential time bias when comparing both groups at the respective

**Fig. 5 | Gene-therapeutic induction of CXCL10 secretion enhances CAR T cell migration and infiltration in vitro. a** Post-enrichment transgene expression rates measured via Q8 positivity in flow cytometry over 8 weeks for transgenic SK-N-AS and SK-N-BE2c cell lines. **b** Supernatant cytokine concentrations for different cytokine-target locus combinations before (pre-sort) and after enrichment (post-sort) for Q8+ cells, determined via ELISA. **c** Flow cytometric analysis of M2 macrophage polarization in response to tumour cell-derived or recombinant IFNG and M-CSF (Macrophage Colony Stimulating Factor). **d** CAR T cell killing dynamics for transgenic and enriched cytokine-expressing SK-N-AS cell lines co-cultivated with either non-transduced T cells or L1CAM-targeting CD4+ and CD8+ CAR T cells at an effector-to-target ratio of 1:5 ($n = 4$ individual co-culture experiments, 2 per donor for CAR T cell production). Killing dynamics were tracked over 96 h via Incucyte live imaging. The dotted line marks 72 h, where statistical comparisons were conducted. Killing at 72 h was determined as the ratio of the t(0) normalized tumour cell count of the treated cell line against the untreated unmodified control. **e** Bioprinted 3D neuroblastoma models were used to analyse CAR T cell infiltration into 3D tumours 12 h post-co-culture. 3D tumour CAR T cell infiltration,

represented by the T cell-to-tumour cell ratio (flow cytometry), was compared between enriched cytokine-expressing and unmodified SK-N-AS and SK-N-BE2c cells. **f** Transendothelial migration and 3D tumour infiltration assays were performed using a HUVEC monolayer with a Boyden transwell insert on a bioprinted 3D neuroblastoma model. Four hours after adding L1CAM-targeting CAR T cells, the insert was removed, and 3D tumour infiltration was measured via flow cytometry after eight hours, following the same procedure as in (**d**). Data presentation: **b**–**f** Mean ± SD. Each dot represents one biological replicate, and exact $n$ values are detailed in the Source Data file. Statistical analysis: **c**, **d** Kruskal−Wallis with Dunn's post hoc test (two-sided, exact $p = 0.021, 0.038, 0.41, 0.13, 0.65, 0.12, 0.4, >0.99, >0.99, >0.99$); **d** two-sided Two-way ANOVA for CAR T cell vs. MOCK comparison (exact $p < 0.0001$); **e**, **f** Mann−Whitney test (two-sided, exact $p = 0.017, 0.0022, 0.029, 0.0095$). $p$ values: *<0.05, **<0.01, ****<0.0001; *n.s.*, not significant. Source data are provided as a Source Data file. Illustration attribution: CAR T cell illustration elements in (**e**, **f**): Created in BioRender. Kuenkele, A. (2025) https://BioRender.com/u71d212.

study endpoint (Fig. 7h). Nevertheless, at endpoint, CXCL10-expressing tumours showed significantly higher infiltration of total human leukocytes ($p = 0.049$) and CAR T cells ($p = 0.01$) (Fig. 7i, and Supplementary Fig. 31), indicating enhanced immune recruitment and CAR T cell trafficking driven by localized CXCL10 expression. Together, these findings demonstrate that targeted CXCL10 knock-in into tumour cells enhances tumoural immune infiltration, improves CAR T cell recruitment, and prolongs tumour control in vivo, supporting the therapeutic potential of tumour-intrinsic cytokine expression for strengthening adoptive cell therapy in solid tumours.

## Discussion

We present a gene-therapeutic strategy that remodels the TME through tumour-intrinsic cytokine expression to enhance immune cell infiltration and immunotherapeutic efficacy, enabled by CancerPAM - a bioinformatics pipeline that identifies patient-specific, tumour-selective CRISPR knock-in target sites. CancerPAM leverages multi-omics datasets to systematically select optimal CRISPR targets, prioritizing specificity and safety while facilitating efficient transgene integration. Applied to data from neuroblastoma samples with a low mutational tumour burden relative to adult malignancies[40,41], CancerPAM identified a median of 130 tumour-specific PAM sites across the entire transcriptome. Using these data, we successfully implemented CRISPR-mediated insertion of pro-inflammatory cytokine genes (CXCL10, CXCL11, IFNG) to enhance immune cell infiltration and function.

CancerPAM's integrative approach, incorporating genomic, transcriptomic and epigenomic data, provides an automated and systematic method to identify tumour-specific PAM sites and minimize off-target risks. The pipeline substantially accelerates the identification process while maintaining high accuracy compared with manual selection. Its ranking algorithm prioritizes sites based on specificity, predicted efficiency and biological relevance to ensure robust therapeutic target selection. Certain limitations remain. Sequencing dataset biases and tumour heterogeneity may influence reproducibility, and the CRISPR efficiency scores used require further improvement. CancerPAM is currently optimized for 5'-NGG-3' PAM sequences (related to the *Streptococcus pyogenes*-derived Cas9 used), limiting compatibility with alternative Cas9 variants. In addition to SNV-derived PAM sites, we also explored structural variant- and ecDNA breakpoints as tumour-specific integration sites outside of the refined CancerPAM pipeline. Although less frequent, breakpoint-derived PAMs and gRNAs further broaden the mutational landscape beyond point mutations. While the feasibility of CRISPR knock-in at ecDNA junctions remains to be determined experimentally, these sites represent intriguing candidates for future studies. However, our analysis revealed that all identified novel targets were highly individualized: only 1.4% of

CancerPAM-identified SNV-PAMs and 5.6% of SV/ecDNA candidates were re-detected in a second patient, and at most a few novel PAMs were shared across up to 7% (4/54) of patients (4 PAMs in 3 patients; 2 PAMs in 4 patients). This low recurrence rate suggests that a broadly shared "tumour-specific safe-harbour" is unlikely to exist in neuroblastoma, at least at the level of SNVs, SVs or ecDNA. Instead, these findings reinforce the rationale for individualized strategies such as CancerPAM, which can identify and rank patient-specific integration sites with high accuracy. The growing feasibility of individualized gene therapies, as highlighted by a recently reported in vivo base-editing therapy in a neonate, further supports the translational potential of personalized targeting approaches guided by CancerPAM[42]. Future pipeline iterations will focus on broadening target recognition (e.g. integrate ecDNA and SV breakpoint analysis), refining ranking criteria and to further integrate functional validation.

CAR T cell recruitment and tumour control in vivo was enhanced through CXCL10 expression by engineered tumour cells. These findings highlight tumour-intrinsic cytokine expression as a promising strategy to reprogram the TME and overcome key barriers to immunotherapy in solid tumours. By enabling localized, sustained secretion of chemokines such as CXCL10 directly from tumour cells, this approach promotes immune cell recruitment and trafficking while avoiding systemic cytokine exposure, potentially minimizing off-target effects and toxicity. Unlike conventional systemic cytokine therapies or exogenous TME modulation, this strategy reprograms the tumour from within, enhancing its susceptibility to immunotherapy. In combination with CAR T cell therapy, we observed improved immune infiltration and tumour control in vitro, in xenografts, and in a humanized NOG (HuNOG) model reconstituted with human CD34+ haematopoietic stem cells, where CXCL10 knock-in increased intratumoural leukocyte and CAR T cell infiltration and prolonged tumour control within a reconstituted human immune microenvironment, supporting this synergistic paradigm.

Although our study focuses on neuroblastoma, the underlying principle may be applicable across a range of immune-excluded solid tumours. In vivo testing was limited to CXCL10 as a representative CXCR3 ligand and candidate, since CXCL10 and CXCL11 share the same receptor and mechanism of action, and stable IFNG knock-in could not be achieved in SK-N-BE2c cells. To highlight the broader relevance of the approach, we compiled a list of additional cytokines, their natural expression in cell line and patient samples and their potential to modulate the TME in neuroblastoma and other solid tumours. These include, for example, CCL19/CCL21, which promote tertiary lymphoid structures and enhance dendritic and T cell recruitment[43,44]; IL-12, IL-15, and IL-18, which drive Th1 polarization and NK/CD8+ T cell activation[45–47]; GM-CSF and Flt3L, which expand and mobilize antigen-presenting cells[48,49]; type I interferons, which

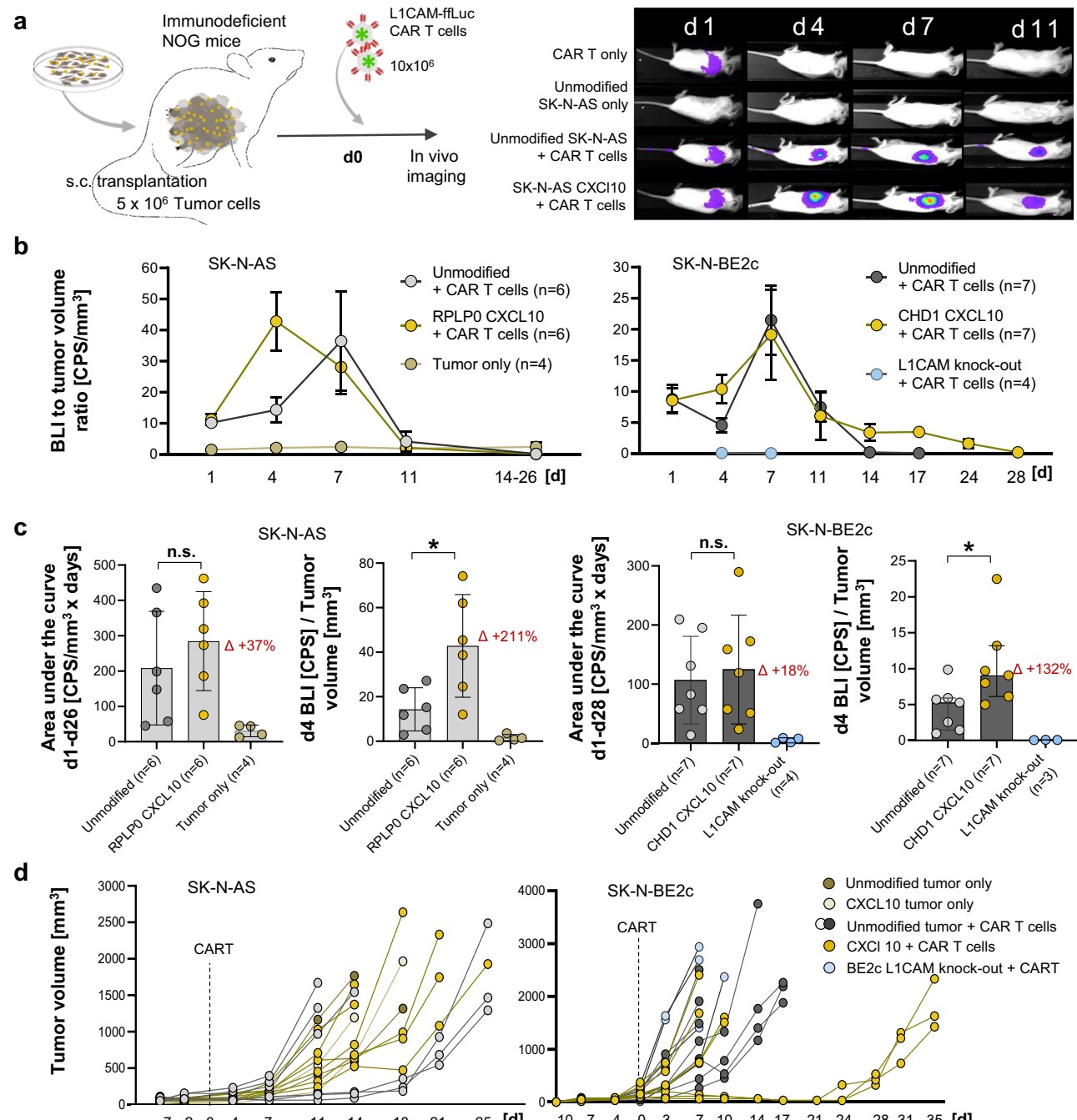

**Fig. 6 | Tumour-intrinsic CXCL10 expression enhances CAR T cell infiltration in xenograft neuroblastoma models. a** Schematic of the xenograft mouse model using immunodeficient NOG (NOD.Cg-Prkdc[scid] Il2rγ[tmISug]/JicTac) mice with subcutaneous transplantation of unmodified SK-N-AS and SK-N-BE2c cells, *RPLP0/CHD1* CXCL10 Q8+ enriched knock-in cell lines, or a SK-N-BE2c L1CAM knock-out cell line in the left flank. After tumour engraftment, mice were treated with firefly luciferase-expressing L1CAM-targeting CD3+ CAR T cells by tail vein injection, followed by repetitive bioluminescence imaging analysis. Representative images of individual animals analysed as groups in (**b**). Images from all animals are provided in Supplementary Figs. 22 and 23. **b** Tumoural CAR T cell infiltration over time, represented as the bioluminescence signal in the left flank relative to the tumour size measured on the same day for different treatment groups. The *SK-N-AS tumour only group*

includes four animals that were excluded from CAR T cell treatment due to tumour sizes smaller than 50 mm³ (Supplementary Fig. 24a). **c** Comparison of total tumoural CAR T cell infiltration and expansion, represented by the area under the curve (AUC) of the bioluminescence-to-tumour volume ratio, and comparison of early infiltration based on the bioluminescence-to-tumour volume ratio on day 4 between treatment groups. **d** Individual tumour growth curves for all treatment groups. Data presentation: **b**, **c** Mean ± SD. Dots in (**b**) represent mean values and dots in (**c**) and *n* represent individual animals. Statistical analysis: (**c**) Mann-Whitney test (two-sided, exact *p* = 0.39, 0.026, >0.99, 0.038). *p* values: *<0.05, **<0.01, ****<0.0001); *n.s.*, not significant. Source data are provided as a Source Data file. Illustration attribution: Elements of tumor cell culture, tumor cell bulk and CAR T cells in (**a**): Created in BioRender. Kuenkele, A. (2025) https://BioRender.com/u71d212.

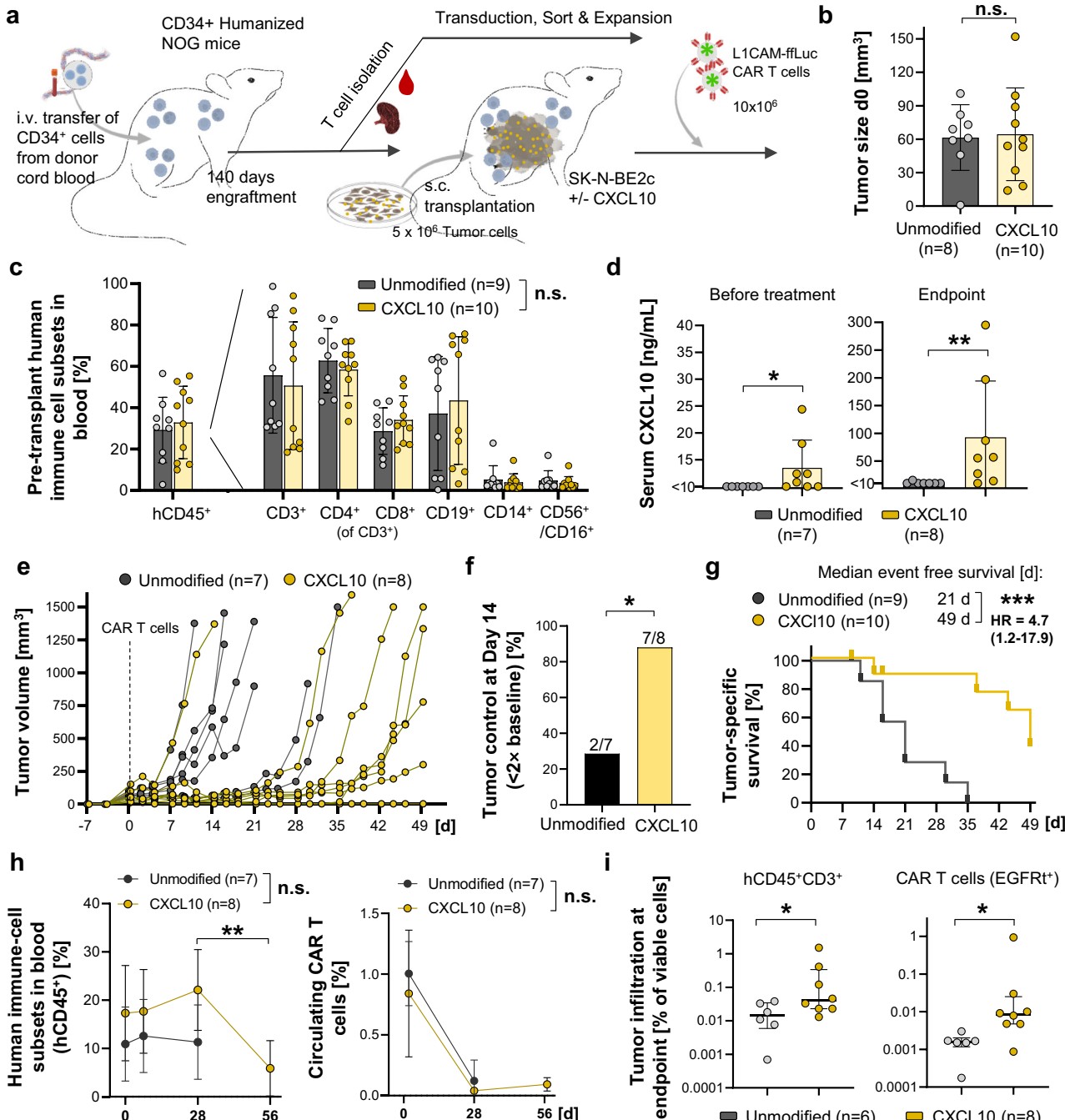

**Fig. 7 | Tumour-intrinsic CXCL10 expression enhances CAR T cell efficacy in humanized CD34+ NOG mice. a** Experimental overview. CD34+ cord-blood-derived human haematopoietic stem cells were engrafted into NOG (NOD.Cg-Prkdc^scid Il2ry^tm1Sug/JicTac) mice to establish a human immune system, followed by collection of T cells for CAR T cell production and subsequent subcutaneous transplantation of SK-N-BE2c neuroblastoma cells (±CXCL10). After engraftment, mice received i.v. luciferase-expressing L1CAM-targeting CAR T cells (10 × 10⁶). **b** Baseline tumour sizes before CAR T cell treatment. **c** Flow-cytometric quantification of peripheral human immune-cell subsets before tumour transplantation (pre-randomization). Human leukocytes are shown as percent of living mononuclear cells; other subsets as percent of human CD45+; CD4+ and CD8+ cells as percent of CD3+ cells. **d** Serum CXCL10 concentrations before treatment and at the experimental endpoint. **e** Individual tumour growth curves after CAR T cell infusion. **f** Percentage of mice with tumour control (≤2 × baseline volume) at day 14. **g** Kaplan–Meier analysis of event-free survival. *n* represents individual animals at time point 0. **h** Proportion of circulating human leukocytes (hCD45+) and CAR T cells (truncated epidermal growth factor receptor tag positive cells (EGFRt+)) in peripheral blood over time. **i** Flow-cytometric quantification of tumour-infiltrating human immune cells (hCD45+/CD3+) and CAR T cells (EGFRt+) at endpoint. Data presentation: **b, c, d, h, i** Mean ± SD. Dots in **h** represent mean values and dots in **b, c, d, i** and *n* represent individual animals. Statistical analysis: **b, d, i** Mann–Whitney test (two-sided, exact *p* = 0.9, 0.026, 0.0044, 0.049, 0.01); **c, h** Two-sided two-way ANOVA for group comparison (exact *p* = 0.66, 0.12); **h** One-Way ANOVA with Friedman test for CXCL10 group (day 28 vs day 56, exact *p* = 0.0056); **f** Fisher's exact test (two-sided, exact *p* = 0.04); **g** log-rank (Mantel–Cox) test (two-sided, exact *p* = 0.0009). *p* values: *<0.05, **<0.01, ***<0.001; n.s., not significant. Source data are provided as a Source Data file. Illustration attribution: Elements of Cord blood stem cells, immune cells, spleen, tumor cell culture, tumor cell bulk and CAR T cells in (**a**): Created in BioRender. Kuenkele, A. (2025) https://BioRender.com/u71d212.

activate dendritic cells and improve antigen cross-priming[50]; and LIGHT (TNFSF14), which normalizes tumour vasculature and facilitates lymphocyte entry[51]. Such candidates underscore that our selection of CXCL10, CXCL11, and IFNG represents a rational proof-of-concept rather than the only possible solution (Supplementary Table 8). Future studies will be needed to evaluate multiplexed or context-dependent cytokine combinations, e.g. further augmented with checkpoint blockade, ideally in optimized immunocompetent preclinical models for neuroblastoma or other entities.

Clinical translation of our strategy depends on improving CRISPR knock-in efficiency and tissue-specific in vivo delivery. Optimized non-viral knock-in approaches, such as modified single-stranded DNA templates and DNA nanostructures have been validated to significantly enhance HDR efficiency[52]. CRISPR delivery strategies like lipid nanoparticles[53], evolved viral particles[54] or peptide-based delivery of CRISPR components[55], show promise for in vivo translation. Particularly, lipid nanoparticle-mediated mRNA delivery has been shown to allow transient CRISPR expression and selective organ-targeting for tissue-specific delivery[53,56,57]. Adeno-associated viral (AAV) vectors, widely used for stable gene delivery, require further refinement to mitigate immunogenicity and insertional mutagenesis risks[58]. Combining AAV and nanoparticle-based approaches could enhance safety and efficiency[59]. Notably, in vivo administration of CAR T cell-encoding mRNA lipid nanoparticles enabled functional CAR T cell production without ex vivo engineering, suggesting a potential adaptation for CRISPR-based TME remodelling to improve immunotherapy in solid tumours[60]. In this context, another important consideration is whether stable genomic integration is needed at all compared with transient delivery of mRNA encoding for TME remodelling cytokines. While mRNA delivery offers flexibility and avoids permanent genomic modification, current platforms lack tumour specificity, provide only short-lived expression, and can trigger innate immune responses that limit efficacy[61–63]. By contrast, genomic integration is technically more complex and less efficient, but once achieved - even at low knock-in frequencies - it enables durable, localized cytokine secretion directly from tumour cells. This continuous production may remodel the TME more effectively than transient pulses, and CancerPAM ensures that expression remains tumour-restricted, thereby enhancing safety. We view both strategies as complementary. The optimal approach will depend on future advances in tumour-specific, non-immunogenic delivery technologies[64,65].

To summarize, our approach - combining CancerPAM-guided CRISPR editing with tumour-intrinsic cytokine expression - offers a versatile and conceptually framework to enhance immunotherapy across solid tumours. By tailoring immune-stimulatory cues within the tumour itself, this strategy can convert immune-excluded tumours into immune-permissive environments, thereby augmenting existing therapies such as CAR T cells or immune checkpoint inhibitors. While our proof-of-concept focuses on neuroblastoma, the platform is broadly applicable to other malignancies with immunosuppressive TMEs. Continued advances in CRISPR delivery technologies and knock-in efficiency will be essential to realizing clinical translation. Together, our findings establish a foundation for precision-engineered TME modulation, highlighting the synergy between computational target selection and therapeutic genome engineering to unlock new avenues in personalized cancer immunotherapy.

## Methods

### Data sources and sample information

Human tumour and matched normal tissue samples ($n = 54$) from patients with neuroblastoma were analysed. The cohort consisted of paediatric and adolescent patients (age range: 1–17 years). Information on participant sex or gender was not available in the anonymised dataset. Sex and gender were not considered as variables in the study design, as there is currently no evidence that these factors influence

the occurrence of CRISPR target sites or affect gene-editing efficacy or safety, and were therefore not relevant to the scientific objectives of this analysis. WGS was performed for patient tumour and matched normal tissue, and expression data were generated from RNA-sequencing (RNA-seq). For comparative in vitro analyses, unpublished data from several neuroblastoma cell lines ($n = 14$) were used. WES and RNA-seq were performed to determine mutational and transcriptional profiles. Extrachromosomal DNA (ecDNA) and structural variants (including deletions, insertions, inversions, tandem duplications and translocations) data were obtained from 20 neuroblastoma primary tumours with matched controls corresponding to the published WGS data of the *discovery cohort*[38]. We used the processed SV calls and ecDNA reconstructions provided by Rodríguez-Fos et al. and deposited with that study[38]; no additional SV calling or ecDNA reconstruction was performed. For downstream PAM analysis, ecDNA coordinates were lifted over to GRCh38 to match the reference genome used in this study[66]. Additionally, publicly available sequencing data from neuroblastoma cell lines ($n = 48$) were included from the Cancer Dependency Map Portal (DepMap 23Q4, 22Q2, 20Q4)[3,67–71]. These datasets comprise mutational profiles generated through high-coverage WES, read-depth analysis and RNA sequencing. Gene dependency scores, gene-expression data, and copy-number variations were also obtained from DepMap. Somatic variants were identified using MuTect2 (version) to call variants between tumour-normal pairs for patients and against the reference genome GRCh38.p14 (release 44) for DepMap and NB cell lines. For comprehensive gene annotation, the reference genome GRCh38.p14 (release 44) was used across all data sets.

### CancerPAM - identification and analysis of novel PAMs

CancerPAM (Python) was designed to process the called variants (csv-files) based on their genome builds (e.g. hg38) and identify somatic variants that produce novel NGG PAMs. The 5' and 3' genomic sequences surrounding the somatic variants were retrieved through an API and the UCSC Genome Browser. This analysis was conducted to assess whether novel C were adjacent to existing C or novel G to existing G. The output included information on the somatic variant, the potential gRNA sequence, the novel PAM and whether the novel PAM was located on the plus or minus strand of the genome. Novel PAMs were analysed for CRISPR efficiency and specificity using CRISPOR[37]. Access to the platform was facilitated through an API, which provided the relevant efficiency and specificity values. CancerPAM was designed using Snakemake to automate and streamline the analysis pipeline[72].

### Analysis of novel PAMs and gRNAs at ecDNA and SV breakpoints

To identify tumour-specific CRISPR-Cas9 target sites in structural variants (SVs) involving extrachromosomal DNA (ecDNA), we developed the bkp-PAM analysis pipeline. This approach builds on previously published neuroblastoma WGS data and breakpoint annotations (Rodriguez-Fos et al.)[38] and extends them with systematic PAM and gRNA discovery at ecDNA/SV-associated junctions.

For each ecDNA/SV breakpoint, we extracted 60 bp windows flanking the junction in the GRCh38 reference genome (−30 to +30 relative to the breakpoint, where the breakpoint lies between positions −1 and +1). Both flanking sequences were strand-corrected according to the reported orientation of the rearrangement breakpoints. For tumour-specific-breakpoint-generated novel PAM site detection, we first assessed whether the breakpoint itself generated a novel CRISPR-Cas9 PAM sequence. Bases at positions −1 and +1 were inspected for the presence of "GG" (NGG PAM motif) or "CC" (CCN motif, reverse complement PAM). To distinguish genuine breakpoint-derived PAMs from pre-existing sites, the flanking sequences were aligned to the wild-type reference genome. A PAM was considered novel if the required G or C was absent in the unaltered genomic context but

introduced by juxtaposition at the breakpoint. For NGG PAMs, the corresponding gRNA sequence was defined as bases −22 to −3 upstream of the PAM; for CCN PAMs, the gRNA was defined as the reverse complement of bases +3 to +22 downstream of the PAM. In addition to de novo PAM creation, we searched for gRNAs spanning the breakpoint that would bind uniquely due to sequence divergence from the reference. Candidate NGG or CCN motifs were identified within ±30 bp around each breakpoint (NGG starting between +6 and +16; CCN starting between −18 and −8). To ensure breakpoint specificity, we required that candidate gRNAs include at least four nucleotides complementary to both sides of the junction. For each breakpoint, the pipeline generated an annotated table with the number and type of novel CRISPR target sites, their genomic coordinates, PAM motif, associated gRNA sequence, and classification as either (a) novel PAM created at the breakpoint or (b) novel gRNA.

## CancerPAM - features and ranking of novel PAMs

In this study, four CRISPOR-derived scores were utilized alongside gene dependency, expression and copy number data. Two scores, MIT and CFD, assess on-target specificity by estimating the guide RNA's genomic uniqueness and considering mismatches with the target DNA. These scores, ranging from 0 to 100, account for the number, position and distribution of mismatches in a sequence-dependent manner, with higher values indicating reduced off-target effects. Notably, the CFD score correlates more strongly with the total off-target cleavage fraction of a guide than the MIT score[5,35,73]. To predict the on-target efficiency the Doench and Moreno-Mateos scores were chosen. The higher the efficiency score, the more likely cleavage occurs at this position. The Moreno-Mateos score is based on CRISPRscan and predicts gRNA activity by analysing molecular features such as guanine enrichment, adenine depletion, nucleotide truncation and 5′ mismatches, effectively capturing the sequence determinants influencing CRISPR/Cas9 activity in vivo[34]. The Doench score, developed from a large-scale CRISPR study, evaluates sequence traits linked to high or low guide RNA activity, creating a scoring algorithm based on desirable nucleotide patterns. These scores also range from 0 to 100[5]. Beside that gene dependency data was used to estimate how essential a gene is for cell survival and the lethality of its knock-out. Highly negative dependency scores indicate critical gene functions, with values below −0.5 signifying depletion in most cell lines and scores below −1 representing strong lethality, corresponding to the median of all essential genes. A value of 0 reflects non-essential genes[3]. To calculate a single score for each novel PAM, they were first ranked for each feature individually. PAMs lacking data for a given feature were assigned the lowest rank for that feature. In the case of ties in feature values, the corresponding PAMs shared the same rank. Subsequently a final score for each novel PAM site was determined by summing the weighted ranks of all features. The Doench and Moreno scores (which assess cutting efficiency but not knock-in efficiency) were averaged. In contrast, the CFD score was assigned a weighting factor of 2, to increase specificity and hence safety. The final score was calculated as follows: 2* rank CFD + rank MIT + 0.5*(rank Doench + rank Moreno) + rank Dependency + rank expression + rank copy number. Scores were then normalized within each dataset to a range of 0–100, where a score of 100 represents the most promising target and a score of 0 the least promising. Finally, PAMs were ranked from highest to lowest score in the output table. In the case of ties the corresponding PAMs shared the same rank.

## Homology-directed repair template design and preparation

Sequences for all homology-directed repair templates (HDRTs) and gRNAs used are provided in Supplementary Table 3. Human cytokine coding sequences were obtained from the NCBI genome viewer (https://www.ncbi.nlm.nih.gov/gdv/). Q8 tag and sPA were used as published[74,75]. DNA fragments were acquired from Twist Bioscience

after codon optimisation. We used homology-based seamless cloning NEBuilder HiFi DNA Assembly, NEB) to insert DNA fragments into a high-copy-number pUC-based plasmid vector, containing a ColE1 origin and Ampicillin resistance (AmpR) before transformation of ultracompetent XL10-Gold (Agilent) *Escherichia coli* cells. Transformed cells were selected on LB agar plates containing ampicillin. Positive clones were identified by colony PCR and their plasmid DNA was isolated using the ZymoPURE Plasmid Miniprep Kit (ZYMO RESEARCH) before Sanger sequencing (LGC Genomics) was conducted for comprehensive analysis. Linear dsDNA HDRTs were produced by PCR amplification using KAPA HiFi DNA Polymerase (Roche) in a 400 µl reaction. Following amplification, the HDRTs were purified using AMPure XP Beads (Beckman Coulter) and their correct size confirmed by gel electrophoresis. For vector expression testing the Effectene transfection reagent (Qiagen) was used in HEK293T cells at a dose of 1 µg DNA per $1 \times 10^6$ cells. Custom genetic constructs and oligonucleotides generated in this study are available from the corresponding author upon reasonable request.

## CRISPR cutting and knock-in

CRISPR cutting and knock-in experiments were conducted using the Lonza 4D-Nucleofector (Lonza, Basel, Switzerland)[76]. Ribonucleoprotein (RNP) complexes were assembled from synthesized single-guide RNAs (gRNAs) (Synthego, Supplementary Table 3) and Alt-R™ S.p. Cas9 Nuclease V3 (IDT) with or without HDRTs and polyethylene glycol (PGA). Electroporation was performed using the X-unit of the 4D-Nucleofector with 16 well Nucleocuvette strips applying pulse programs DN110 for SK-N-BE2c and FF104 for SK-N-AS cells. For CRISPR-Cas knock-in experiments, the RNP complex was assembled immediately before nucleofection. Per reaction, a gRNA to Cas9 protein molar ratio of 2:1 was used (80 pmol gRNA (2.5 µg) and 40 pmol Cas9 enzyme (6.67 µg) per $1 \times 10^6$ cells) with 50 µg polyethylene glycol (PGA) were mixed with sterile water. The complex was incubated at room temperature for 15 min. HDRT dsDNA (2.0 µg per $1 \times 10^6$ tumour cells and 1.0 µg per $1 \times 10^6$ T cells) was then added. Tumour cells were trypsinized, counted, washed twice ($200 \times g$, 5 min, room temperature) and resuspended in SF buffer containing Supplement 1 (1:4.5 dilution). T cells were resuspended in supplement containing buffer P3. For each reaction, 20 µL of the cell suspension was added to the corresponding RNP-HDRT mixture and electroporation was carried out using Lonza 4D-Nucleofector 16-well strips. Following nucleofection, cells were recovered in antibiotic-free RPMI medium supplemented with 10% FBS, incubated at 37 °C with 5% $CO_2$ and transferred to appropriate cell culture plates. For T cell knock-ins Alt-R HDR Enhancer V2 (IDT) at 1 µM final concentration was added. CRISPR knock-in experiments targeting tumour cells did not include an Alt-R HDR Enhancer V2 (IDT). After 24 h, culture conditions were adjusted as necessary, including the removal of HDR Enhancer (IDT) when applicable. For CRISPR editing efficiency assessment, custom PCR assays were designed using the GeneGlobe tool (Qiagen) to amplify the cut sites. Out/Out PCR reactions were performed using the QIAprep & CRISPR Kit (Qiagen) with AllTaq Master Mix. Sanger sequencing was conducted and the resulting chromatograms were analysed using the ICE (Synthego) and EditR (http://baseeditr.com/) tools to quantify indel formation.

## Digital PCR for copy number variation analysis

To assess the efficiency and stability of genetic modifications in transgenic cell lines, digital PCR (dPCR) was performed using the QIAcuity digital PCR system (QIAGEN). Primers and hydrolysis probes were designed based on thermodynamic and structural parameters optimized for the QIAcuity digital PCR system. GC content (40–60%), primer melting temperature (55–65 °C), and probe Tm (1–3 °C higher than primers) were adjusted to avoid secondary structures and cross-dimerization. Specificity was confirmed by NCBI BLAST. All primer and probe sequences are available in Supplementary Table 4. Primers were

synthesized by Eurofins Genomics and hydrolysis probes by Metabion. Copy number variation (CNV) analysis was conducted using a duplex probe-based approach to quantify the integrated transgenes relative to a reference gene. Genomic DNA was extracted from stable transgenic and wild-type control cell lines. The dPCR reaction mix contained the QIAcuity Probe PCR Master Mix (QIAGEN), gene-specific primers and probes and fragmented template DNA. The total reaction volume was 12 μl for a 96-well nanoplate format and 40 μl for a 24-well format. The final reaction composition included: 1× QIAcuity Probe PCR Master Mix, 800 nM forward and reverse primers for the target transgene, 400 nM for the reference gene (AFF3) primers, 400 nM target-specific hydrolysis probe (FAM-labelled) and 200 nM AFF3 probe (HEX-labelled). To optimize template accessibility, 0.05 U/μl XbaI (New England Biolabs) was added. Samples were pipetted into the QIAcuity Nanoplate (QIAGEN) and sealed with a QIAcuity Nanoplate Seal (QIAGEN). The dPCR thermal cycling protocol was performed on the QIAcuity One digital PCR system (QIAGEN) with the following conditions for In/In transgene-specific assays: Enzyme activation at 95 °C for 2 min, denaturation at 95 °C for 15 s, primer annealing and elongation at 58 °C for 30 s. This cycle was repeated for 40 cycles, followed by an imaging step for fluorescent signal detection. For Out/In site-specific knock-in assays annealing at 5 8 °C for 30 s was followed by an additional elongation step for 1 min at 72 °C and cycle count was increased to 55 cycles followed by a 2-min extension at 72 °C to enhance signal resolution. Following amplification, the QIAcuity software (QIAGEN) analysed partitioned fluorescence signals to determine absolute DNA copy numbers per microliter using Poisson statistics. Each sample was processed as a duplex reaction, normalizing transgene copy number to the AFF3 reference to account for variations in DNA input and to calculate cell counts. The software generated graphical representations of positive and negative partitions, scatterplots and quantitative tables. Negative control wells (no-template controls) were included to validate specificity and fluorescence thresholding was manually reviewed to ensure correct partition classification. Final CNV values were expressed as copies per 100 cells.

## CAR T cell generation

CAR T cells were generated from PBMCs isolated from the blood of healthy donors using a density gradient centrifugation method[18]. Blood was diluted 1:1 with phosphate-buffered saline (PBS) and carefully layered over Ficoll-Paque (Sigma-Aldrich) in a centrifuge tube. Following centrifugation at $300 \times g$ for 20 min without brake, the mononuclear cell layer was collected. Red blood cells were lysed using a haemolysis buffer, and the remaining cells were washed with PBS and counted. T cells were isolated from PBMCs the CD3+ Pan-T Cell Isolation Kit (Miltenyi), followed by magnetic-activated cell sorting (MACS) as per the manufacturer's instructions. Isolated T cells were seeded at a concentration of $1 \times 10^6$ in a 24-well plate and activated with anti-CD3/CD28 beads at a 1:1 cell-to-bead ratio. Depending on the CAR construct, T cells were transduced with lentiviral vectors (SIN epHIV7) propagated in 293T cells on Day 1 post-activation at a multiplicity of infection of 1. non-transduced T cells served as negative controls. Transduced T cells were cultured in T cell medium (RPMI + 10% FBS + 1% GlutaMAX (Gibco) supplemented with 0.5 ng/ml interleukin-15 (IL-15, Miltenyi)) and 5 ng/ml interleukin-7 (IL-7, Miltenyi), with medium and cytokine replenishment every 2–3 days. Twelve days post-transduction, transduction efficiency was assessed by immunostaining for epidermal growth factor receptor truncated (EGFRt) and analysed by flow cytometry. EGFRt-positive cells were enriched using MACS. Cells were stained with a PE-labelled anti-EGFRt antibody, followed by incubation with magnetic anti-PE beads and subsequently separated magnetically. The enriched T cells were then cryopreserved for future use. Prior to experimental use for in vitro experiments, cryopreserved CAR T cells were thawed and subjected to an expansion protocol. For in vivo experiments, CAR T cells were enriched using MACS, expanded according to the rapid expansion protocol, and cryopreserved until use. This involved co-culturing the CAR T cells with freshly thawed, irradiated (80 Gy) PBMCs and irradiated (35 Gy) EBV-transformed lymphoblastoid feeder cells (TM-LCL), in the presence of OKT3 (Miltenyi) CD3 activating antibody complex, IL-7 and IL-15. The culture medium was refreshed every 2–3 days with the addition of 0.5 ng/ml IL-15 and 5 ng/ml IL-7 for in vitro and NOG xenograft experiments and 10 ng/ml IL-15 and 10 ng/ml IL-7 for humanized NOG (HuNOG) experiments. Experiments were conducted 12–15 days after the initiation of the second stimulation.

## 3D bioprinted tumour infiltration model

To assess CAR T cell infiltration into 3D neuroblastoma tumours, bioprinted tumour models were generated[39]. Bioprinting was performed by Cellbricks GmbH (Berlin, Germany) using a standardized hydrogel-based bioink optimized for neuroblastoma culture, allowing the creation of standardized cylindrical tumour constructs embedding SK-N-AS and SK-N-BE2c neuroblastoma cells (±cytokine knock-in) with precise size and volume (Ø 3 mm × 1 mm; 7.07 mm³), enabling controlled cell distribution and tumour architecture. L1CAM-targeting CAR T cells were then co-cultured with the 3D tumours for 12 h and infiltration efficiency was quantified by flow cytometry, using the T cell-to-tumour cell ratio as a readout. To further model physiological barriers to T cell migration, a transendothelial migration and tumour infiltration assay was developed by incorporating a Human Umbilical Vein Endothelial Cells (HUVEC) monolayer within a Boyden transwell insert, positioned atop the bioprinted tumour constructs. This system simulated vascular endothelium, requiring CAR T cells to migrate through an endothelial barrier before reaching the tumour mass. Vybrant™ DiO Cell-Labelling Solution (Thermo Fisher, Cat# V22886) was used to confirm HUVEC layer formation before T cell addition. After 4 h of CAR T cell migration, the insert was removed, and CAR T cell infiltration into the tumour mass was assessed 8 h later by flow cytometry.

## Housing and handling of animals

Mice were housed under specific pathogen-free conditions in individually ventilated cages with a 12 h light/dark cycle, at an ambient temperature of 23 ± 1 °C and relative humidity of 45–65%, with ad libitum access to food and water. The animal welfare was checked twice daily. Body weights, tumour volume and general health conditions were recorded throughout the whole study.

## Xenograft mouse model, in vivo CAR T cell transplantation and bioluminescence imaging

All animal experiments were conducted in female NOD.Cg-Prkdc<scid> Il2ry<tm1Sug>/JicTac mice (CIEA NOG; Taconic Biosciences, Inc.) aged 6–8 weeks at the start of the experiment. A total of 24 mice were used for humanization and 48 mice for non-humanized xenograft CAR T-cell experiments. Only female mice were used to minimize variability in tumour growth kinetics and immune reconstitution related to sex hormones; therefore, sex was not included as an experimental variable in the analyses. For transplantation $5 \times 10^6$ tumour cells were mixed with Matrigel (1:1) and transplanted subcutaneously in a final volume of 100 μl in the right flank. After engraftment (palpable tumour), tumour size was measured at least twice a week with a digital caliper. Tumour volume was calculated with the formula $V = (L \times W^2)/2$. CAR T cell transplantation was performed at a tumour volume of at least 50 mm³. For in vivo experiments, $10 \times 10^6$ CAR T cells were injected into the tail vein (i.v.) in 200 μl PBS by slow injection. For BLI mice were anaesthetised with Isoflurane (Baxter, San Juan, Puerto Rico) and received intraperitoneally 150 mg/kg D-Luciferin (Biosynth, Staad, Switzerland) dissolved in PBS. BLI was performed with the NightOwl II LB983 in vivo imaging system. The IndiGO 2.0.5.0 software was used for initial analysis, colour-coding of the signal intensity and quantification. BLI was

performed d1, d4, d7, d11 and d14-26 for SK-N-AS and d1, d4, d7, d11, d17, d24 and d28 for SK-N-BE2c transplanted animals. All mice treated in this study were sacrificed upon reaching a tumour volume of 1.5 cm³, which represents the maximal tumour size permitted by the institutional animal ethics approval (LaGeSo, Reg E0023-23), or earlier if other humane endpoints were met. All animals were euthanized before exceeding this limit. Blood samples were collected after retro-bulbar venous plexus puncture in MiniCollect® tubes containing Lithium Heparin for serum samples and processed according to the manufacturer's instruction. Serum samples were stored at −80 °C. For tumour tissue collection, mice were sacrificed. Tumours were removed, their weights determined and subsequently divided into two pieces, one of which was formalin-fixed and paraffin-embedded and one as a snap-frozen sample.

**Humanized NOG (HuNOG) mouse model and engraftment.** Human umbilical cord blood and buffy coat samples were obtained under informed consent and isolation of human CD34+ haematopoietic stem cells (HSCs) and their transplantation into NOG mice were performed at EPO Berlin-Buch. HSCs were isolated from umbilical cord blood or buffy coat samples by Ficoll density gradient separation followed by MACS using CD34 microbeads (Miltenyi Biotec), as previously described[77]. Purity of CD34+ cells was confirmed by flow cytometry. HSCs were cryopreserved in freezing medium (10% DMSO, 90% FBS) and stored in liquid nitrogen until use. Prior to transplantation, samples were thawed in 50% FBS/50% PBS, counted, and resuspended in 200 μl PBS for intravenous injection. Recipient NOG mice were irradiated (1 Gy) 4–24 h before HSC transplantation. A total of 24 mice were engrafted with HSCs derived from five independent donor pools (groups A-E; $n = 4$–6 per group). One animal per group was sacrificed post-engraftment for spleen harvest and CD3+ T-cell isolation to generate CAR T cell products. For each donor pool, peripheral blood from all remaining animals was pooled with the spleen sample to increase total T-cell yield for CAR T cell manufacturing. Autologous CAR T cells were successfully produced for groups A and B, which subsequently received autologous infusions, while the remaining animals received allogeneic CAR T cells from donor groups A or B in a balanced design. CAR T cell production failed for three donor pools due to one instance of bacterial contamination and two cases with insufficient CD3+ T-cell yield. Immune reconstitution was confirmed by flow cytometry 140 days post-CD34+ stem cell transplantation prior to tumour xeno-grafting and CAR T cell infusion.

**Statistics & reproducibility.** Sample sizes were based on prior experience with comparable experiments and reflect standards commonly used in the field. Given the exploratory nature of this study and the limited availability of primary samples and animals, no formal statistical method was used to predetermine sample size. Data were excluded only from animals that met pre-defined exclusion criteria, such as pre-infusion graft-versus-host disease, engraftment failure or ectopic tumour localisation as detailed in the Results section. For in vitro experiments, data were excluded only when technical errors occurred (e.g. electroporation failure, cell viability <70%, or flow cytometry data with <5000 gated events). No other data were excluded from analyses. Animal treatment allocation and analysis were randomized and performed in a blinded manner. In vitro experiments were randomized with respect to treatment allocation and measurement order, but investigators were not blinded during these assays. Biological replicates were defined as independent experiments performed on separate cell preparations, animals, or donor-derived samples. For in vitro experiments, one biological replicate corresponds to an independent electroporation, transfection, or co-culture performed on a separate cell passage or donor. Technical replicates (e.g. replicate wells or repeated measurements) were averaged within each biological replicate and not used for statistical testing. For in vivo

experiments, one biological replicate corresponds to a single animal. Exact n values for each condition are provided in the figures, figure legends, or Source Data file. All statistical analyses were performed using GraphPad Prism (version 10.2.0). Data are presented as mean ± standard deviation (SD) unless stated otherwise. For comparisons of more than two groups, Kruskal–Wallis tests with Dunn's post hoc correction were applied. Pairwise comparisons were conducted using Mann-Whitney tests. Two-way analysis of variance (ANOVA) with Tukey's post hoc test was used for multiple comparisons across conditions and time points. Linear regression was applied for curve fitting, while Spearman's rank correlation was used for correlation analyses, as it is more robust for small sample sizes where $R^2$ values are expected to be low. Logistic regression was used for growth curve fitting and growth coefficients k were used for proliferation rate comparison ($Y = (Y_m \cdot Y_0) / [(Y_m - Y_0) \cdot \exp(-k \cdot x) + Y_0]$). For survival analysis in in vivo studies, Kaplan–Meier curves were generated. Bioluminescence to tumour volume ratios and total CAR T cell infiltration over time were analysed using AUC calculations followed by Mann-Whitney tests. For dPCR-based quantifications, two-way ANOVA was applied. All statistical tests were two-sided unless stated otherwise. Multiple comparisons were adjusted as indicated (Dunn's, Tukey's). Statistical significance was defined as $p < 0.05$ (p values: *<0.05, **<0.01, ***<0.001, ****<0.0001; n.s., not significant). Exact p values of key analyses from in are given in the figure legends, all other p values are available in the source data file. Sample sizes for each experiment are indicated in the respective figures and figure legends, as well as in the source data file.

Additional methods are described in the supplementary methods section. Equipment, consumables, antibodies and software used are listed in Supplementary Tables 5–7.

### Ethics statement

All experiments were performed in accordance with relevant ethical regulations for research involving human participants and animals. Collection and reuse of human tumour and matched normal tissue samples ($n = 54$) were conducted under written informed consent and approved by the Ethics Committee of Charité - Universitätsmedizin Berlin (protocol EA2/055/17), in accordance with the Declaration of Helsinki and applicable data protection laws. The generation of CAR T cells was conducted under the ethical approval EA2/262/20 from Charité - Universitätsmedizin Berlin. Human umbilical cord blood and buffy coat samples used for generation of CD34+ haematopoietic stem cells were obtained under informed consent and processed at EPO Berlin-Buch in compliance with institutional and national regulations. No participant compensation was provided for sample donation. All animal procedures (humanized and non-humanized xenograft NOG models) were approved by the Landesamt für Gesundheit und Soziales (LaGeSo), Berlin, Germany (approval Reg E0023-23) and performed in compliance with German animal-welfare legislation. Mice were housed under standardized conditions with veterinary oversight, and humane endpoints were pre-defined and observed.

### Reporting summary

Further information on research design is available in the Nature Portfolio Reporting Summary linked to this article.

## Data availability

*Raw sequencing data from patient samples.* The whole-exome and transcriptome sequencing data generated in this study from patient tumour and matched healthy tissue samples ($n = 54$) have been deposited in the German Human Genome-Phenome Archive (GHGA) under the accession code GHGAS41175626365361. These data are available under restricted access due to data privacy regulations and ethical requirements related to personal data from a vulnerable patient population. Access to the data is granted only to qualified researchers for non-commercial research use upon approval of a controlled access

request. Requests for data access can be submitted via the GHGA Data Portal [https://data.ghga.de] or by contacting ppk-c@charite.de. Applications are reviewed by the Data Access Committee (DAC) at Charité - Universitätsmedizin Berlin in coordination with GHGA, and responses and data access are typically provided within 3–4 weeks. Approved applicants are required to sign a Data Use Agreement (DUA) governed by Charité - Universitätsmedizin Berlin. A copy of the DUA is available upon request from the corresponding author or from the DAC. Users of these data are requested to cite this study in any resulting publications. Data reuse is limited to non-commercial research purposes. The raw whole-genome sequencing data from 20 neuroblastoma primary tumours with matched controls used to call structural variants and reconstruct extrachromosomal DNA (ecDNA) elements in this study are publicly available in the European Genome–Phenome Archive (EGA) under accession numbers: EGAS00001001308 [https://ega-archive.org/studies/EGAS0000 1001308], EGAS00001004022, and EGAS00001006983. The processed structural variant calls and ecDNA reconstructions supporting this study's conclusions are publicly available in Zenodo under https://doi.org/10.5281/zenodo.8032024, corresponding to the published whole-genome sequencing dataset of the discovery cohort by Rodríguez-Fos et al., (Cell Genomics, 2023)[78].

*Cell line sequencing data.* Sequencing data generated from neuroblastoma cell lines in this study have been deposited in the NCBI Sequence Read Archive (SRA) under BioProject accession number PRJNA1292401. The datasets are publicly available and can be accessed directly through the project webpage or by entering the accession number in the NCBI SRA Run Selector.

*DepMap datasets.* Publicly available sequencing and dependency datasets from the Cancer Dependency Map (DepMap) were used in this study:

DepMap 23Q4 [https://doi.org/10.25452/figshare.plus.246679 05.v2][69];

DepMap 22Q2 [https://doi.org/10.6084/m9.figshare.197000 56.v2][68].

DepMap 20Q4 [https://doi.org/10.6084/m9.figshare.132370 76.v4][67];

*Other data.* All other data supporting the findings of this study are available within the article or its Supplementary Information. Source data are provided with this paper.

## Code availability

The CancerPAM bioinformatics pipeline used in this study, including sample data, is available in the Code Ocean repository under accession number 7671597 under https://doi.org/10.24433/CO.2312035.v1[79]. The ecDNA/SV Breakpoint-CRISPR pipeline is available in the Zenodo repository under accession number 17209179 under https://doi.org/10.5281/zenodo.17209179[80]. The Python implementation of the CancerPAM statistical tumour off-target risk model is available in the Zenodo repository under accession number 17209430 under https://doi.org/10.5281/zenodo.17209430[81].

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

## Acknowledgements

This work was supported by Berliner Krebsgesellschaft (grant number LAFF202008 to M.L.) and KINDerLEBEN e.V. Berlin (to M.L.). Additional support was provided by Charité - Universitätsmedizin Berlin and the Berlin Institute of Health (BIH) at Charité - Universitätsmedizin Berlin (AdHoc Booster Grant to D.W. and M.L.). M.L. participates in the BIH Charité Clinician Scientist Program, funded by Charité - Universitätsmedizin Berlin and the BIH. A.K. participates in the BIH Charité Advanced Clinician Scientist Pilot Program, also funded by Charité - Universitätsmedizin Berlin and the BIH. Parts of this research were carried out within the scientific framework of the DFG-funded Collaborative Research Center CRC1588 (493872418). K.T. was supported as a junior clinician scientist within Collaborative Research Center CRC1588 (493872418). The authors thank Silke Schwiebert and Anika Winkler for their technical support with the experiments; Michael C. Jensen for providing the L1CAM-CAR T cell construct; Nikolaus Rajewsky for his support as a scientific mentor within the framework of the BIH Charité Clinician Scientist Program; and Kathy Astrahantseff for manuscript proofreading and editorial advice. Parts of Figs. 1, 5, 6, 7 and Supplementary Figs. 16 and 19 were created with BioRender.com.

## Author contributions

M.L. conceived and designed the study, acquired funding, and wrote the manuscript with input from all authors. M.L., J.H., M.L.A. and M.J. developed the CancerPAM pipeline. M.L., J.M., S.A., S.B., M.P., E.O., C.V., C.L., L.A., K.T., M.L.A., J.K. and M.S. performed the experiments. M.L., J.M., S.A., J.H., S.B., M.P., E.O. and C.V. analysed the data. M.L., M.J., C.V., L.A., K.T., F.Z., J.K., M.S., E.R., K.A., D.L.W., A.G.H., R.K. and A.K. developed the methodology. RK, AE and AK supervised the study. All authors contributed equitably to the work and were given the opportunity to participate in data analysis, interpretation, and manuscript preparation. The collaboration involves institutions from multiple countries and disciplines, emphasising transparency, inclusivity, and equitable recognition of scientific contributions.

## Funding

## Competing interests

M.L. reports in-kind support (reagents and services) related to CRISPR-Cas gene editing and digital PCR from QIAGEN. This support did not influence the design, execution, analysis, or interpretation of the present study. D.L.W.'s laboratory at Charité has received in-kind reagents/services related to CRISPR–Cas gene editing from Integrated DNA Technologies (IDT) and GenScript Inc. None of the companies or intellectual property described influenced the design, execution, or interpretation of this study. Unrelated to this work, D.L.W. is named as an inventor on patent applications related to genome editing and cell therapies and is a co-founder of TCBalance Biopharmaceuticals GmbH. All other authors declare no competing interests.

## Additional information

¹Department of Pediatric Oncology and Haematology, Charité - Universitätsmedizin Berlin, corporate member of Freie Universität Berlin, Humboldt Universität zu Berlin, Berlin, Germany. ²Berlin Institute of Health at Charité - Universitätsmedizin Berlin, Berlin, Germany. ³German Cancer Consortium (DKTK), partner site Berlin and German Cancer Research Center (DKFZ), Heidelberg, Germany. ⁴Berlin Center for Advanced Therapies (BeCAT), Charité - Universitätsmedizin Berlin, Berlin, Germany. ⁵Berlin Institute for Medical Systems Biology (BIMSB), Max Delbrück Center for Molecular Medicine in the Helmholtz Association, Berlin, Germany. ⁶Cellbricks GmbH, Berlin, Germany. ⁷BIH Center for Regenerative Therapies (BCRT), Charité - Universitätsmedizin Berlin, Berlin, Germany. ⁸EPO Berlin-Buch GmbH, Berlin, Germany. ⁹Experimental and Clinical Research Center (ECRC) of the MDC and Charité - Universitätsmedizin Berlin, Berlin, Germany. ¹⁰Institute of Medical Immunology, Charité - Universitätsmedizin Berlin, Berlin, Germany. ¹¹Center for Cell and Gene Therapy, Baylor College of Medicine, Houston, TX, USA. ¹²Department of Molecular and Cellular Biology, Baylor College of Medicine, Houston, TX, USA. ¹³Dan L Duncan Comprehensive Cancer Center, Baylor College of Medicine, Houston, TX, USA. ¹⁴Max Delbrück Center for Molecular Medicine in the Helmholtz Association (MDC), Berlin, Germany. ¹⁵University Medicine Essen, Essen, Germany. ✉e-mail: michael.launspach@charite.de

