## [Transparent Peer Review file · Nature Communications]

Personalized CRISPR Knock-In Cytokine Gene Therapy to Remodel the Tumor Microenvironment and Enhance CAR T Cell Therapy in Solid Tumors

Corresponding Author: Dr Michael Launspach

Version 0:

Reviewer comments:

Reviewer #1

(Remarks to the Author)

In the manuscript, the authors tried to apply CancerPAM to patient tumor and cell line sequencing data to identify optimal integration sites for pro-inflammatory cytokines which reprogram the tumor microenvironment for increasing immune cell infiltration and immunotherapy response. The following issues should be addressed more clearly.

1. Since a lot of inflammatory cytokines play the roles in modulating immune response, the rational in the study for selecting cytokines (CXCL10, CXCL11, IFNG) should be addressed. Additional experiments should be performed.
2. Since tumor contains a lot of mutations, how to avoid unspecific knock-in in the system?
3. All the three settings, the results showed that only tumor-secreted CXCL10 could increase CAR T cell migration and infiltration in the animal model. Why not CXCL11 and IFNG? What are the underlying mechanisms?
4. For the animal analysis, the NOG model is not enough. The system should be validated using the wild-type mice.

(Remarks on code availability)

Reviewer #2

(Remarks to the Author)

SUMMARY

Launspach et al. present a novel CRISPR-based strategy to remodel the tumor microenvironment (TME) by integrating cytokine genes into tumor cells using tumor-specific novel PAM sites identified via CancerPAM, a custom multi-omics bioinformatics pipeline. The approach combines precision knock-in of CXCL10, CXCL11, or IFNG into neuroblastoma cells and demonstrates improved CAR T cell infiltration and modest tumor control in vitro and in xenografted mice. The study integrates computational target selection with functional validation, showing proof-of-principle for tumor-intrinsic immunomodulation as a strategy to overcome the immunosuppressive TME in solid tumors.

ASSESSMENT

This is an impressive and well-written manuscript with both conceptual novelty and thorough experimental execution. The authors convincingly demonstrate that CancerPAM enables the identification of tumor-specific CRISPR knock-in sites and that targeted insertion of chemokine-encoding genes enhances immune cell migration and infiltration. The work is methodologically rigorous, and the breadth of validation (in vitro, 3D models, and in vivo) strengthens its translational relevance. Overall, this is a great manuscript and a good fit for Nature Communications. That said, I have a couple of questions regarding the potential scalability of the approach and potential ideas for alternative approaches.

MAJOR

1. CancerPAM is a great idea, but maybe the authors could discuss a little more what the benefit of such a personalized integration would be vs. safe-harbor integration in the tumor? My comment is all about scalability. HDR is tough, and while

you can tune it in this case for a specific patient/tumor, it also slows things down and makes them more complex/costly. You show in Fig. 5g that cytokine measurements look actually somewhat higher in AAVS1 cytokine-integration cells? Would that translate into higher killing, too? It would be great to compare that side-by-side with a CancerPAM outcome in a normal in vitro killing system (like Fig. 5c). Sorry if I missed this.

2. Could the authors please discuss why the integration of a cytokine gene is conceptually important and not just mRNA delivery to tumor cells? Maybe they can try mRNA delivery in a simple in vitro setting to demonstrate the functionality and further expand/broaden the concept? This would greatly reduce complexity and enhance scalability. I do get the point about tumor-specific integration but what's the harm if a TME cell itself expressed the cytokine? In the end, one should potentially discuss in how far this is not rather a delivery challenge, i.e. how to get the mRNA into the tumor.

3. Have you seen known fusion genes in the neuroblastoma cell lines you use? Maybe those could be used as tumor-specific safe-harbors that are at least somewhat more often "shared" by different patients?

4. It is known that extrachromosomal DNA (ecDNA) plays a substantial role in neuroblastoma (and in many other cancers), as far as I know. Have you detected ecDNA in your cells/samples? Would those be a potential target for integration as well, given they are somewhat tumor-specific and also contain plenty of oncogenes? Just an idea.

MINOR

5. Line 174: pipeline demonstrated 99% accuracy >> please explain.

6. Fig. 3e/4g: Would it make sense to show R2 for correlation in these scatter plots?

(Remarks on code availability)

Version 1:

Reviewer comments:

Reviewer #1

(Remarks to the Author)

The authors have addressed most of my concerns.

(Remarks on code availability)

Reviewer #2

(Remarks to the Author)

Thanks a lot to the authors for addressing all of my concerns. Congratulations on this great manuscript!

(Remarks on code availability)

Response to reviewers for manuscript

“Personalized CRISPR knock-in cytokine gene therapy to remodel the tumor microenvironment and enhance CAR T cell therapy in solid tumors”

We thank the reviewers for their thoughtful and constructive feedback, which has helped us substantially improve the clarity, rigor, and overall quality of the manuscript, and which we have addressed in detail as outlined below.

New additions or substantial changes in the manuscript are highlighted in yellow. Edits made to improve language or reduce word count are highlighted in grey. To enhance readability and comply with journal length recommendations, we moved several methodological sections to the Supplementary Methods, including Cell Lines and Cell Culture, Cytotoxicity Assays, Macrophage Polarization Assay, and Statistical Estimation of Tumor-Specific CRISPR Off-Target Risk.

Reviewer #1:

In the manuscript, the authors tried to apply CancerPAM to patient tumor and cell line sequencing data to identify optimal integration sites for pro-inflammatory cytokines which reprogram the tumor microenvironment for increasing immune cell infiltration and immunotherapy response. The following issues should be addressed more clearly.

1. Since a lot of inflammatory cytokines play the roles in modulating immune response, the rationale in the study for selecting cytokines (CXCL10, CXCL11, IFNG) should be addressed. Additional experiments should be performed.

We thank the reviewer for highlighting the importance of clarifying our rationale for selecting CXCL10, CXCL11, and IFNG. We have now substantially expanded the Introduction to explicitly address this point. Neuroblastoma is characterized by an immunologically “cold” tumor microenvironment (TME) with sparse cytotoxic immune cell infiltration, transcriptional repression of interferon-stimulated genes, impaired antigen presentation machinery, and exclusion of CD8⁺ T cells and NK cells, particularly in MYCN-amplified tumors. Low interferon- γ (IFNG) signatures and reduced expression of CXCL9/10 are consistently associated with poor prognosis and decreased immune responsiveness in neuroblastoma. Conversely, tumors with T cell-inflamed gene signatures, including IFNG and its downstream chemokines, demonstrate improved survival. Introducing pro-inflammatory cytokines such as CXCL10, CXCL11, and IFNG therefore directly addresses the fundamental barrier of immune exclusion. Mechanistically, CXCL10 and CXCL11 are ligands for CXCR3, expressed on activated CD8⁺ T cells and NK cells (and on CAR T cells, as confirmed in our study), and their tumor expression has been associated with improved immune infiltration and therapeutic outcomes across multiple preclinical and clinical settings. IFNG, a master regulator of antitumor immunity, induces these chemokines, enhances MHC expression, and improves antigen presentation. We have integrated these points with supporting references to the revised Introduction.

We agree with the reviewer that additional functional evidence for IFNG was warranted, given that much of our original focus was on CAR T cell migration. In response, we have performed two new sets of experiments that are now included in the Results: (I) Tumor cell intrinsic IFNG signaling: Flow cytometry analysis confirmed that tumor cell-targeted IFNG expression promoted upregulation of HLA-ABC (MHC class I), FAS, and PD-L1, consistent with enhanced antigen presentation capacity and immunoregulatory effects. (II) Macrophage polarization assay: Using donor-derived human monocytes

in a transwell co-culture, we observed that IFNG expression by tumor cells significantly inhibited M2-like macrophage polarization. This aligns with prior reports that tumor-associated macrophages (TAMs) in neuroblastoma strongly contribute to immune suppression and disease progression [1–4,7]. Our data therefore demonstrate that IFNG expression not only influences tumor-intrinsic signaling but also impacts the myeloid compartment. These new data further support the rationale for including IFNG as a candidate cytokine and are described in the revised Results section.

We furthermore agree that many additional cytokines may be of interest for tumor-intrinsic immune remodeling. Our study was designed as a proof-of-concept to demonstrate the feasibility of CancerPAM using well-characterized candidates (CXCL10, CXCL11, IFNG), not to claim these are the most effective or universal. To address this point, we added a supplementary table summarizing other promising cytokines with relevance for neuroblastoma and solid tumors (**Supplementary Table 8**). This table includes a summary of expression of these cytokines in neuroblastoma cell lines and patient data from our study which we visualized and added as supplement figures (**Supplementary Fig. 34 & 35**). These data demonstrate that neither CXCL10, CXCL11, nor IFNG are expressed in neuroblastoma cell lines or patient tumor samples, thereby further supporting the rationale for selecting these candidates. Other examples include the following: CCL5 and CXCL9 cooperate with CXCL10 to orchestrate effector T cell trafficking, though CCL5 can also recruit Tregs and MDSCs in some contexts; CCL19/CCL21 organize tertiary lymphoid structures and enhance dendritic and T cell recruitment; GM-CSF and Flt3L expand antigen-presenting cells and dendritic subsets; type I interferons, which activate dendritic cells and improve antigen cross-priming, and Th1 cytokines such as IL-12, IL-15, and IL-18 potently activate NK and CD8⁺ T cells. Moreover, LIGHT (TNFSF14) normalizes tumor vasculature and induces high endothelial venules, thereby facilitating T cell infiltration. We have now included these examples with references in the revised Discussion and supplementary material, emphasizing that future work should evaluate such cytokines - alone or in rational combinations- in optimized preclinical models for neuroblastoma.

2. Since tumor contains a lot of mutations, how to avoid unspecific knock-in in the system?

We thank the reviewer for raising this important point. We agree that tumor genomes harbor numerous mutations, and in principle, tumor-specific off-target integration events could occur. Such events may arise if (1) a second novel PAM site exists with high similarity to the on-target guide sequence or (2) a gRNA with several mismatches in the reference genome is “rescued” to higher similarity by the presence of a tumor-specific mutation. Currently, the CancerPAM pipeline does not explicitly predict such tumor-specific off-targets. However, for any gRNA designed through the pipeline, it is straightforward to perform post hoc testing against both the tumor and germline genome to exclude guides with high off-target potential prior to use. Additionally, non-specific knock-in at additional tumor-specific loci would in most cases be functionally neutral or even beneficial (by increasing the proportion of modified tumor cells), and would only be problematic in rare instances where integration disrupts a bona fide tumor suppressor gene.

Furthermore, the risk of tumor-specific off-targets depends mainly on two factors: (a) the total number of somatic variants in the tumor and (b) the number of genomic sequences with partial similarity (≤ 4 mismatches) to the chosen gRNA which is reflected in the established CRISPR specificity scores (CFD, MIT) and which are integral components of the CancerPAM ranking. Thus, higher CancerPAM scores intrinsically correlate with a lower risk of tumor-specific off-target sites. To support this, we developed a complementary analysis tool that calculates the statistical probability of tumor-specific off-targets

based on the number of somatic variants and gRNA sequence similarity for any chosen gRNA. When applied to representative gRNAs identified via CancerPAM from patient samples, this analysis confirmed that CancerPAM ranking strongly correlates with reduced off-target risk, with 95% of top-ranked gRNAs (ranks 1–3) showing a calculated risk of <0.01%. This correlation analysis has been added to **Supplementary Fig. S13** and the results section.

Taken together, these analyses demonstrate that while tumor-specific off-targets are theoretically possible, the CancerPAM pipeline substantially mitigates this risk by prioritizing high-specificity gRNAs, and additional per-guide checks can easily be performed prior to final selection.

3. All the three settings, the results showed that only tumor-secreted CXCL10 could increase CAR T cell migration and infiltration in the animal model. Why not CXCL11 and IFNG? What are the underlying mechanisms?

We thank the reviewer for this comment. We would like to clarify that CXCL11 and IFNG were not tested in the 3D bioprinted tumor model or in in vivo xenograft experiments; only CXCL10 was evaluated in these settings. This was partly due to practical considerations of time and resources, and partly because our study was designed as a proof-of-concept to establish the feasibility of tumor-intrinsic cytokine expression for enhancing CAR T cell function. Also, CXCL10 and CXCL11 both signal through the CXCR3 receptor, which is highly expressed on CAR T cells, CD8⁺ T cells, and NK cells. Given their shared mechanism of action, we selected CXCL10 as a representative ligand for in vivo testing. Furthermore, stable knock-in of IFNG could not be achieved in SK-N-BE2c cells due to IFNG-associated growth impairment and cell loss, making it technically unfeasible to include IFNG in xenograft experiments for both cell lines used. We therefore emphasize that the absence of CXCL11 and IFNG in the in vivo setting does not imply a lack of efficacy, but rather reflects the scope of this initial proof-of-concept study. Future work will extend these analyses to additional cytokines, including CXCL11 and IFNG, to systematically compare their relative contributions to tumor remodeling and CAR T cell activity.

4. For the animal analysis, the NOG model is not enough. The system should be validated using the wild-type mice.

We thank the reviewer for this important comment. We fully agree that experiments in immunocompetent wild-type mice would provide complementary insights. However, this was not feasible within the scope of the current revision. Establishing such a system would have required development of an entirely new murine platform, including CAR T cells specific for a murine neuroblastoma antigen, knock-in of murine cytokines into murine neuroblastoma cell lines, and repeating all in vitro validation studies in the murine context. Given the extensive time and resources required, this was beyond the scope of the present study.

Additionally, we note that there is robust evidence that murine immune systems do not fully mimic human immune responses, particularly in cytokine signaling and immune cell trafficking, which has led to the increased preference for humanized mouse models over wild-type mice in predictive in vivo studies. Key differences include species-specific cytokine and chemokine receptor expression, divergent leukocyte subset distributions, and distinct signaling pathway components, all of which impact immune cell development, activation, and migration [1-3]. For example, more than 80 major immunological differences have been catalogued between mice and humans, including variations in cytokine profiles (e.g., IL-2, IL-4, IFN- γ), chemokine receptor expression, and immune cell trafficking mechanisms. These

discrepancies contribute to the frequent failure of murine models to predict human therapeutic responses, especially in the context of inflammatory and autoimmune diseases [1-4]. Consequently, humanized mouse models - immunodeficient mice engrafted with human hematopoietic cells or tissues - are increasingly utilized to study human-specific immune responses, drug efficacy, and disease mechanisms. These models better recapitulate human immune cell development, trafficking, and cytokine signaling, providing a more predictive platform for preclinical research [4-8]. However, challenges remain in fully reconstructing human immune function, and ongoing efforts focus on improving these models to enhance their translational relevance [6-8].

Hence, to regardless strengthen our *in vivo* analysis, we performed an additional experiment using humanized CD34⁺ HuNOG mice reconstituted with a human immune system and xenografted with SK-N-BE2c neuroblastoma cells with or without CXCL10 knock-in at the CHD1 locus as an alternative. Following infusion of L1CAM-targeting CAR T cells, CXCL10-expressing tumors exhibited markedly slower growth, higher tumor control rates at day 14 (88 % vs. 29 %, $p = 0.04$), and significantly prolonged median event-free survival (49 vs. 21 days; $p = 0.0009$; HR = 4.7, 95 % CI 1.2–17.9) compared to controls (**new Fig. 7e–g, Supplementary Fig. S30**). Flow-cytometric analysis confirmed increased intratumoral infiltration of human leukocytes ($p = 0.049$) and CAR T cells ($p = 0.01$) in CXCL10-expressing tumors (**Fig. 7i, Supplementary Fig. S31**). While CAR T cell bioluminescence tracking was not quantifiable in this model (**Supplementary Fig. S33 & Supplementary Note 15**), functional validation demonstrated intact luciferase activity and CAR T cell persistence in blood and tumor tissue. These data provide robust evidence that tumor-intrinsic CXCL10 expression enhances immune and CAR T cell recruitment and improves therapeutic efficacy within a reconstituted human immune microenvironment.

Taken together, while we acknowledge that experiments in wild-type mice would provide complementary information, our additional HuNOG data better capture the complexity of human immune interactions than the simple NOG xenograft model and further support the potential of tumor-intrinsic cytokine expression to enhance CAR T cell efficacy *in vivo*. To the best of our knowledge, this is also the first study to employ a humanized CD34⁺ HuNOG model to assess CAR T cell efficacy in neuroblastoma. While humanized mice have been used for CAR T research in other solid tumors, we are not aware of prior work applying this strategy specifically to neuroblastoma with tumor-engineered cytokine expression. This adds further novelty to our study and represents an important step toward developing more appropriate models for combinatorial immunotherapy approaches.

1. *Of Mice and Not Men: Differences Between Mouse and Human Immunology.* Mestas J, Hughes CC. *Journal of Immunology (Baltimore, Md. : 1950).* 2004;172(5):2731-8. doi:10.4049/jimmunol.172.5.2731.

2. *Differences in Innate Immune Response Between Man and Mouse.* Zschaler J, Schlorke D, Arnhold J. *Critical Reviews in Immunology.* 2014;34(5):433-54.

3. *A Comprehensive Atlas of Immunological Differences Between Humans, Mice, and Non-Human Primates.* Bjornson-Hooper ZB, Fragiadakis GK, Spitzer MH, et al. *Frontiers in Immunology.* 2022;13:867015. doi:10.3389/fimmu.2022.867015.

4. *Use of Humanized Mice to Study the Pathogenesis of Autoimmune and Inflammatory Diseases.* Koboziev I, Jones-Hall Y, Valentine JF, et al. *Inflammatory Bowel Diseases.* 2015;21(7):1652-73. doi:10.1097/MIB.0000000000000446.

5. *Overcoming Current Limitations in Humanized Mouse Research.* Brehm MA, Shultz LD, Luban J, Greiner DL. *The Journal of Infectious Diseases.* 2013;208 Suppl 2:S125-30. doi:10.1093/infdis/jit319.
6. *Human Immune Responses and Potential for Vaccine Assessment in Humanized Mice.* Akkina R. *Current Opinion in Immunology.* 2013;25(3):403-9. doi:10.1016/j.coi.2013.03.009.
7. *The Development and Improvement of Immunodeficient Mice and Humanized Immune System Mouse Models.* Chen J, Liao S, Xiao Z, et al. *Frontiers in Immunology.* 2022;13:1007579. doi:10.3389/fimmu.2022.1007579.
8. *Humanized Mice for Immune System Investigation: Progress, Promise and Challenges.* Shultz LD, Brehm MA, Garcia-Martinez JV, Greiner DL. *Nature Reviews. Immunology.* 2012;12(11):786-98. doi:10.1038/nri3311.

Reviewer #2:

SUMMARY

Launspach et al. present a novel CRISPR-based strategy to remodel the tumor microenvironment (TME) by integrating cytokine genes into tumor cells using tumor-specific novel PAM sites identified via CancerPAM, a custom multi-omics bioinformatics pipeline. The approach combines precision knock-in of CXCL10, CXCL11, or IFNG into neuroblastoma cells and demonstrates improved CAR T cell infiltration and modest tumor control in vitro and in xenografted mice. The study integrates computational target selection with functional validation, showing proof-of-principle for tumor-intrinsic immunomodulation as a strategy to overcome the immunosuppressive TME in solid tumors.

ASSESSMENT

This is an impressive and well-written manuscript with both conceptual novelty and thorough experimental execution. The authors convincingly demonstrate that CancerPAM enables the identification of tumor-specific CRISPR knock-in sites and that targeted insertion of chemokine-encoding genes enhances immune cell migration and infiltration. The work is methodologically rigorous, and the breadth of validation (in vitro, 3D models, and in vivo) strengthens its translational relevance. Overall, this is a great manuscript and a good fit for Nature Communications. That said, I have a couple of questions regarding the potential scalability of the approach and potential ideas for alternative approaches.

MAJOR

1. CancerPAM is a great idea, but maybe the authors could discuss a little more what the benefit of such a personalized integration would be vs. safe-harbor integration in the tumor? My comment is all about scalability. HDR is tough, and while you can tune it in this case for a specific patient/tumor, it also slows things down and makes them more complex/costly. You show in Fig. 5g that cytokine measurements look actually somewhat higher in AAVS1 cytokine-integration cells? Would that translate into higher killing, too? It would be great to compare that side-by-side with a CancerPAM outcome in a normal in vitro killing system (like Fig. 5c). Sorry if I missed this.

3. Have you seen known fusion genes in the neuroblastoma cell lines you use? Maybe those could be used as tumor-specific safe-harbors that are at least somewhat more often “shared” by different patients?

4. It is known that extrachromosomal DNA (ecDNA) plays a substantial role in neuroblastoma (and in many other cancers), as far as I know. Have you detected ecDNA in your cells/samples? Would those be a potential target for integration as well, given they are somewhat tumor-specific and also contain plenty of oncogenes? Just an idea.

We thank the reviewer for raising these three important points regarding safe-harbor integration versus patient-specific tumor targeting. First, it is important to differentiate between (1) general “safe-harbors” such as AAVS1 and (2) tumor-specific recurrent safe-harbors (e.g. recurrent SNPs, fusion genes or tumor-specific ecDNA sequences) that could be exploited across multiple patients. In our SNP analysis we did not identify recurrent tumor-specific safe-harbors suitable for a generalized approach.

General safe-harbor loci such as AAVS1 allow robust integration but are not tumor-specific. Without a delivery system that restricts integration to tumor cells, this carries a substantial safety concern, as cytokine knock-in would also occur in healthy tissues. By contrast, targeting tumor-specific sites

identified by CancerPAM ensures integration is confined to tumor cells, vastly improving safety even if integration efficiency may be somewhat lower. Knock-in efficiency and cytokine expression were highest at the AAVS1 locus, for which our HDR template was optimized and which is biallelic. The co-culture killing assays we originally presented were performed with AAVS1 knock-in cells; we have clarified this in the revised manuscript. To directly address this point, we performed additional co-culture experiments using IFNG-expressing neuroblastoma cells with knock-in at AAVS1 versus the tumor-specific RPLP0 locus in SK-N-AS cells (low L1CAM target expression). Both pre-sort and enriched conditions showed significantly enhanced killing within the first 24h of co-culture, but no significant difference between AAVS1 and RPLP0 integration. Notably, even non-enriched knock-in cells showed this effect, underscoring that low-frequency integration can still have measurable biological consequences by remodeling the TME. In SK-N-BE2c cells, stable IFNG knock-in could not be achieved; therefore, we modeled dose-dependency by adding recombinant IFNG at varying concentrations. Here, no additional IFNG-mediated effect on killing was observed, most likely because the high L1CAM antigen abundance drove rapid CAR T cell killing that masked more subtle cytokine effects. These new data are included in the revised Results section and **Supplementary Fig. S26**. Taken together, these experiments indicate that even at relatively low-frequency, tumor-specific knock-ins can provide meaningful TME remodeling effects without requiring maximum expression levels achievable at a general safe-harbor. We therefore argue that prioritizing safety (tumor-specific knock-in) over absolute integration efficiency (general safe-harbor) is justified. Finally, we fully agree with the reviewer that a tumor-specific recurrent safe-harbor locus would represent an optimal solution for scalability, as it could be pre-established and fine-tuned across patients. To explore this possibility, we initiated a collaboration with Anton Henssen's laboratory, which provided breakpoint-level structural variant (SV) and extrachromosomal DNA (ecDNA) data from primary neuroblastoma patient samples. Using these datasets, we performed a computational analysis to identify novel PAM sites and gRNA sequences absent from the reference genome and screened for recurrence across patients. We identified unique tumor-specific PAMs and gRNAs at individual SV breakpoints and ecDNA junctions (**now included in Figure 2b**), which we now discuss as potential candidates for follow-up knock-in studies. Notably, it remains unknown whether CRISPR knock-in strategies are effective for ecDNA targets. However, only 5.6% of novel PAMs/gRNAs from ecDNA/SVs were detected again in a second patient, and we did not find recurring candidates shared across multiple patients. These findings suggest that the genomic heterogeneity of neuroblastoma - at the levels of SNPs, SVs, and ecDNA - may limit the feasibility of defining broadly shared "tumor-specific safe-harbors." Instead, our results reinforce the rationale for individualized strategies such as CancerPAM, which can identify patient-specific integration sites. We would also like to highlight that patient-individualized gene therapies are becoming increasingly feasible with streamlined platform technologies. For example, the first patient-specific in vivo base-editing therapy for a neonate with carbamoyl-phosphate synthetase 1 deficiency was reported this year in the *New England Journal of Medicine* (Musunuru et al., NEJM 2025), underscoring the emerging translational potential of individualized editing approaches. We have incorporated a discussion on target site recurrence and patient-specific versus safe-harbor targeting approaches in the revised manuscript.

2. Could the authors please discuss why the integration of a cytokine gene is conceptually important and not just mRNA delivery to tumor cells? Maybe they can try mRNA delivery in a simple in vitro setting to demonstrate the functionality and further expand/broaden the concept? This would greatly reduce complexity and enhance scalability. I do get the point about tumor-specific integration but what's the harm if a TME cell itself expressed the cytokine? In the end, one should potentially discuss in how far

this is not rather a delivery challenge, i.e. how to get the mRNA into the tumor.

This is a very important point. We agree that mRNA delivery to tumors is a promising alternative strategy, and we have expanded the Discussion to address this aspect. Both approaches have distinct advantages and disadvantages:

mRNA delivery: The transient nature of mRNA expression, flexible dosing, and avoidance of permanent genomic modification are well-established advantages. However, current mRNA delivery platforms, particularly lipid nanoparticles, lack tumor specificity and often result in off-target expression, notably in the liver, which complicates the maintenance of therapeutic cytokine levels and increases the risk of systemic toxicity [9-11]. Moreover, while transient mRNA expression can increase safety, sustained expression in solid tumors remains a challenge due to inefficient delivery and rapid degradation, while the intrinsic immunogenicity of mRNA - mediated by activation of innate immune sensors such as toll-like receptors (TLRs), RIG-I, and MDA5 - can limit transgene expression and provoke inflammation (also outside of tumors), especially with repeated dosing. These limitations necessitate extensive nucleotide modification and purification to mitigate immunogenicity and improve safety [12-14].

Stable genomic integration: Stable genomic integration via homology-directed repair (HDR) is conceptually more complex and less efficient, with efficiency and specificity varying by locus and tumor biology [15-18]. This approach necessitates carefully designed HDR templates and optimized delivery strategies, thereby increasing technical complexity and cost. The risk of insertional mutagenesis persists if off-target integration occurs, though specificity ranking algorithms such as CancerPAM can mitigate this risk [16,17]. Unlike transient mRNA approaches, stable integration is permanent, which may be disadvantageous if cytokine expression becomes toxic or needs rapid shut-off. Nevertheless, even low-frequency knock-in enables durable, localized cytokine secretion, potentially remodeling the tumor microenvironment more effectively than transient mRNA pulses, especially when tumor-restricted expression is achieved [15,16].

With respect to the reviewer's suggestion of an additional experiment, we would like to clarify that we are actively pursuing mRNA delivery to neuroblastoma in a separate project, and a dedicated manuscript describing first results in methods development is currently under review elsewhere. Including preliminary data in the present manuscript would dilute its scope, which is specifically focused on the conceptual and technical feasibility of tumor-intrinsic knock-in via CancerPAM. We therefore chose to address this point by extending the Discussion rather than adding experimental mRNA data. *For reviewer reference only*, we provide preliminary results demonstrating the general feasibility of (co-) delivery of DNA and mRNA to the neuroblastoma cell lines SK-N-AS and SK-N-BE2c using lipid nanoparticles (**Review letter figure 1**).

[Figure Redacted]

Finally, we concur with the reviewer that this is fundamentally a delivery challenge. If tumor-specific, safe, and non-immunogenic mRNA delivery technologies became available, they could indeed represent a highly attractive alternative or complement to stable integration. At present, however, integration via CancerPAM provides a conceptual framework for safe and potentially more durable cytokine expression in tumors. We have extended our Discussion to reflect these considerations.

9. *mRNA-based Cancer Therapeutics*. Liu C, Shi Q, Huang X, et al. *Nature Reviews. Cancer*. 2023;23(8):526-543. doi:10.1038/s41568-023-00586-2.

10. *Applications of mRNA Delivery in Cancer Immunotherapy*. Pan X, Zhang YW, Dai C, et al. *International Journal of Nanomedicine*. 2025;20:3339-3361. doi:10.2147/IJN.S500520.

11. *mRNA Delivery Systems for Cancer Immunotherapy: Lipid Nanoparticles and Beyond*. Estapé Senti M, García Del Valle L, Schiffelers RM. *Advanced Drug Delivery Reviews*. 2024;206:115190. doi:10.1016/j.addr.2024.115190.

12. *mRNA Therapeutics in Cancer Immunotherapy*. Beck JD, Reidenbach D, Salomon N, et al. *Molecular Cancer*. 2021;20(1):69. doi:10.1186/s12943-021-01348-0.

13. *Advances in mRNA Therapeutics for Cancer Immunotherapy: From Modification to Delivery*. Han G, Noh D, Lee H, et al. *Advanced Drug Delivery Reviews*. 2023;199:114973. doi:10.1016/j.addr.2023.114973.

14. *Nanoparticle-Based Delivery Strategies of Multifaceted Immunomodulatory RNA for Cancer Immunotherapy*. Yoo YJ, Lee CH, Park SH, Lim YT.

Journal of Controlled Release : Official Journal of the Controlled Release Society. 2022;343:564-583. doi:10.1016/j.jconrel.2022.01.047.

15. *Engineering Tripartite Gene Editing Machinery for Highly Efficient Non-Viral Targeted Genome Integration.* Nam H, Xie K, Majumdar I, et al. *Nature Communications.* 2025;16(1):4569. doi:10.1038/s41467-025-59790-3.

16. *Novel Extragenic Genomic Safe Harbors for Precise Therapeutic T-Cell Engineering.* Odak A, Yuan H, Feucht J, et al. *Blood.* 2023;141(22):2698-2712. doi:10.1182/blood.2022018924.

17. *New Human Chromosomal Sites With "Safe Harbor" Potential for Targeted Transgene Insertion.* Pellenz S, Phelps M, Tang W, et al. *Human Gene Therapy.* 2019;30(7):814-828. doi:10.1089/hum.2018.169.

MINOR

5. Line 174: pipeline demonstrated 99% accuracy >> please explain.

We thank the reviewer for pointing out that this statement required clarification. The reported 99% accuracy refers to the concordance between CancerPAM's automated ranking and manual curation of novel PAM sites from whole-exome sequencing data. In benchmarking, CancerPAM correctly identified 99% of tumor-specific PAM sites detected by manual screening, validating the reliability of the computational system. We have revised the Results section to clarify that this accuracy metric represents concordance with manual curation.

6. Fig. 3e/4g: Would it make sense to show R² for correlation in these scatter plots?

We thank the reviewer for this suggestion. We note, however, that given the small number of data points in these analyses, R² values are expected to be low and may not be fully informative. For this reason, we primarily relied on non-parametric Spearman correlation analyses, which to the best of our knowledge are more robust in this context and less dependent on assumptions of linearity. Nevertheless, to increase transparency, we included R² values in the revised figures (**Fig. 3e, Fig. 4g/h**) and added a note in the methods section that Spearman correlation was preferred for correlation analysis.

We greatly appreciate the reviewers' time and effort and remain happy to clarify or discuss any remaining aspects further.